# A chimeric nuclease substitutes a phage CRISPR-Cas system to provide sequence-specific immunity against subviral parasites

Zachary K Barth[1†], Maria HT Nguyen[1], Kimberley D Seed[1,2]*

[1]Department of Plant and Microbial Biology, University of California, Berkeley, Berkeley, United States; [2]Chan Zuckerberg Biohub, San Francisco, United States

**Abstract** Mobile genetic elements, elements that can move horizontally between genomes, have profound effects on their host's fitness. The *phage-inducible chromosomal island-like element* (PLE) is a mobile element that integrates into the chromosome of *Vibrio cholerae* and parasitizes the bacteriophage ICP1 to move between cells. This parasitism by PLE is such that it abolishes the production of ICP1 progeny and provides a defensive boon to the host cell population. In response to the severe parasitism imposed by PLE, ICP1 has acquired an adaptive CRISPR-Cas system that targets the PLE genome during infection. However, ICP1 isolates that naturally lack CRISPR-Cas are still able to overcome certain PLE variants, and the mechanism of this immunity against PLE has thus far remained unknown. Here, we show that ICP1 isolates that lack CRISPR-Cas encode an endonuclease in the same locus, and that the endonuclease provides ICP1 with immunity to a subset of PLEs. Further analysis shows that this endonuclease is of chimeric origin, incorporating a DNA-binding domain that is highly similar to some PLE replication origin-binding proteins. This similarity allows the endonuclease to bind and cleave PLE origins of replication. The endonuclease appears to exert considerable selective pressure on PLEs and may drive PLE replication module swapping and origin restructuring as mechanisms of escape. This work demonstrates that new genome defense systems can arise through domain shuffling and provides a greater understanding of the evolutionary forces driving genome modularity and temporal succession in mobile elements.

*For correspondence:
kseed@berkeley.edu

Present address: †Department of Microbiology, Cornell University, Ithaca, United States

## Introduction

Mobile genetic elements (MGEs), genetic units capable of spreading within and between genomes, are key mediators of evolution. MGEs differ vastly in size and complexity. At one end of the spectrum are homing endonucleases (HEGs). These single-gene MGEs occur in diploid loci and cleave cognate alleles so that their coding sequences can serve as templates for recombinational repair (*Stoddard, 2011*). At another extreme, integrative viruses are complex MGEs that can encode their own replication genes as well as structural components for their own dispersal (*Krupovic et al., 2019*), and even cargo genes dedicated to boosting the fitness of their cellular hosts (*Harrison and Brockhurst, 2017*). While the evolutionary importance of MGEs across domains of life is clear, apart from a handful of exceptions (*Greenwood et al., 2018*), it is not possible to study the spread of MGEs in real time in populations of multicellular organisms. In contrast, short generation times and low barriers to horizontal gene transfer make bacteria ideal organisms for studying how MGEs shape the evolution of their hosts.

Recent work has led to a deeper appreciation of the extensive entanglement between MGEs and genome defense functions. MGEs frequently encode defense modules to prevent infection by viruses or other nonvirus MGEs (*Koonin et al., 2020*). Such modules include toxin-antitoxin (TA)

systems, restriction modification (RM) systems, CRISPR-Cas, and numerous other systems recently described (*Doron et al., 2018*; *Gao et al., 2020*; *Koonin and Makarova, 2019*; *Marraffini, 2015*; *Millman et al., 2020*; *Mruk and Kobayashi, 2014*). Beyond their antiviral and anti-MGE functions, defense systems also serve the selfish needs of the MGEs that encode them, and the constituents of defense modules can be recruited to benefit viruses and nonvirus MGEs by serving counter-defense functions. Viruses may encode antitoxin or DNA modification genes as a means of escaping TA and RM systems endogenous to their hosts (*Loenen and Raleigh, 2014*; *Otsuka and Yonesaki, 2012*). While many defense modules eliminate invading viruses and MGEs through nucleolytic attack, many phages use nucleases to degrade the host genome, preventing further expression of host defenses (*McKitterick et al., 2019a*; *Panayotatos and Fontaine, 1985*; *Parson and Snustad, 1975*; *Souther et al., 1972*; *Warner et al., 1975*).

The flow of MGEs and their defense systems between viruses and hosts, as well as the retooling of genes for defense, counter-defense, and MGE maintenance or dispersal functions, has been described using the model 'guns for hire' to reflect the mercenary nature of these defense systems and the selfish MGEs that carry them (*Koonin et al., 2020*). One of the most compelling examples of host-pathogen conflicts that conforms to the 'guns for hire' framework occurs in *Vibrio cholerae* between the bacteriophage ICP1 and its own parasite, the *p*hage-inducible chromosomal island-*l*ike element (PLE) (*Seed et al., 2013*). Upon infection by ICP1, PLE excises from the host chromosome, replicates to high copy (*O'Hara et al., 2017*), and is assembled into transducing particles to spread the PLE genome to new cells (*Netter et al., 2021*). PLE excision and DNA replication both require ICP1-encoded gene products (*Barth et al., 2020b*; *McKitterick et al., 2019a*; *McKitterick and Seed, 2018*). Similarly, multiple lines of evidence including shared host cell receptors (*O'Hara et al., 2017*), PLE genome analysis, and electron microscopy (*Netter et al., 2021*) strongly suggest that PLE is packaged into remodeled ICP1 virions for mobilization. While infected PLE(+) cells still die, no ICP1 virions are produced when PLE activity is unimpeded (*O'Hara et al., 2017*). Thus, PLE prevents further spread of ICP1 and protects the host cell population. In this way, PLE acts as both a selfish parasite of ICP1 and an effective abortive infection defense system for *V. cholerae*.

True to the 'guns for hire' model, ICP1 has co-opted a genome defense system to protect itself from PLE. Many ICP1 isolates encode a CRISPR-Cas system that can destroy PLE within the infected cell and restore ICP1 reproduction (*Seed et al., 2013*; *Figure 1A*). CRISPR-Cas systems are typically adaptive immune systems that provide immunological memory against specific nucleic acid sequences (*Barrangou et al., 2007*; *Marraffini, 2015*). The memory function of CRISPR-Cas is achieved through the integration of 'spacers,' short DNA sequences derived from viruses or MGEs, that are integrated into an array of spacer repeats. The spacer can then be transcribed to serve as an RNA guide that directs nucleolytic machinery against complementary sequence. In this way, acquisition of a small portion of foreign DNA provides the specificity required for defense.

Reflecting the primary role of ICP1's CRISPR-Cas as an anti-PLE system, almost all spacers associated with the system are PLE derived (*Seed et al., 2013*; *McKitterick et al., 2019b*). Like cellular CRISPR-Cas systems, the ICP1 system can acquire new immunological memory, reflecting that PLE is not a single static genome but that a number of PLE variants exist. To date, five PLE variants, numbered 1–5, have been described, occurring in about ~15% of sequenced epidemic *V. cholerae* genomes. There is a pattern of temporal succession, where one PLE will dominate in sequenced genomes for a time before being supplanted by another PLE (*O'Hara et al., 2017*), but the reemergence of old PLE sequence in new PLE variants suggests that unsampled reservoirs exist in nature. Like PLEs, there is also diversity among ICP1 genomes. Not all ICP1 isolates encode CRISPR-Cas, but this does not mean that they are defenseless against PLEs. Previous work found that an ICP1 variant that naturally lacked CRISPR-Cas was able to reproduce on the two oldest PLE variants, PLE5 and PLE4, as well as the most recent variant PLE1 (*O'Hara et al., 2017*). Much like PLE variants, there appears to be some temporal succession in the presence or absence of ICP1's CRISPR-Cas system. A minority of ICP1 isolates collected between 2001 and 2011 possessed CRISPR-Cas systems (*Angermeyer et al., 2018*), while CRISPR-Cas encoding ICP1 predominated between 2011 and 2017 (*McKitterick et al., 2019b*). As PLE and ICP1 have coevolved specific mechanisms of parasitism and counter-defenses, it is worth exploring if the temporal succession of PLE and ICP1 variants could be in response to selective pressures that the two entities exert on each other.

Intrigued by CRISPR-independent interference of PLE and hoping to gain insight into patterns of ICP1 and PLE variant succession, we set out to identify the mechanism of PLE interference in ICP1

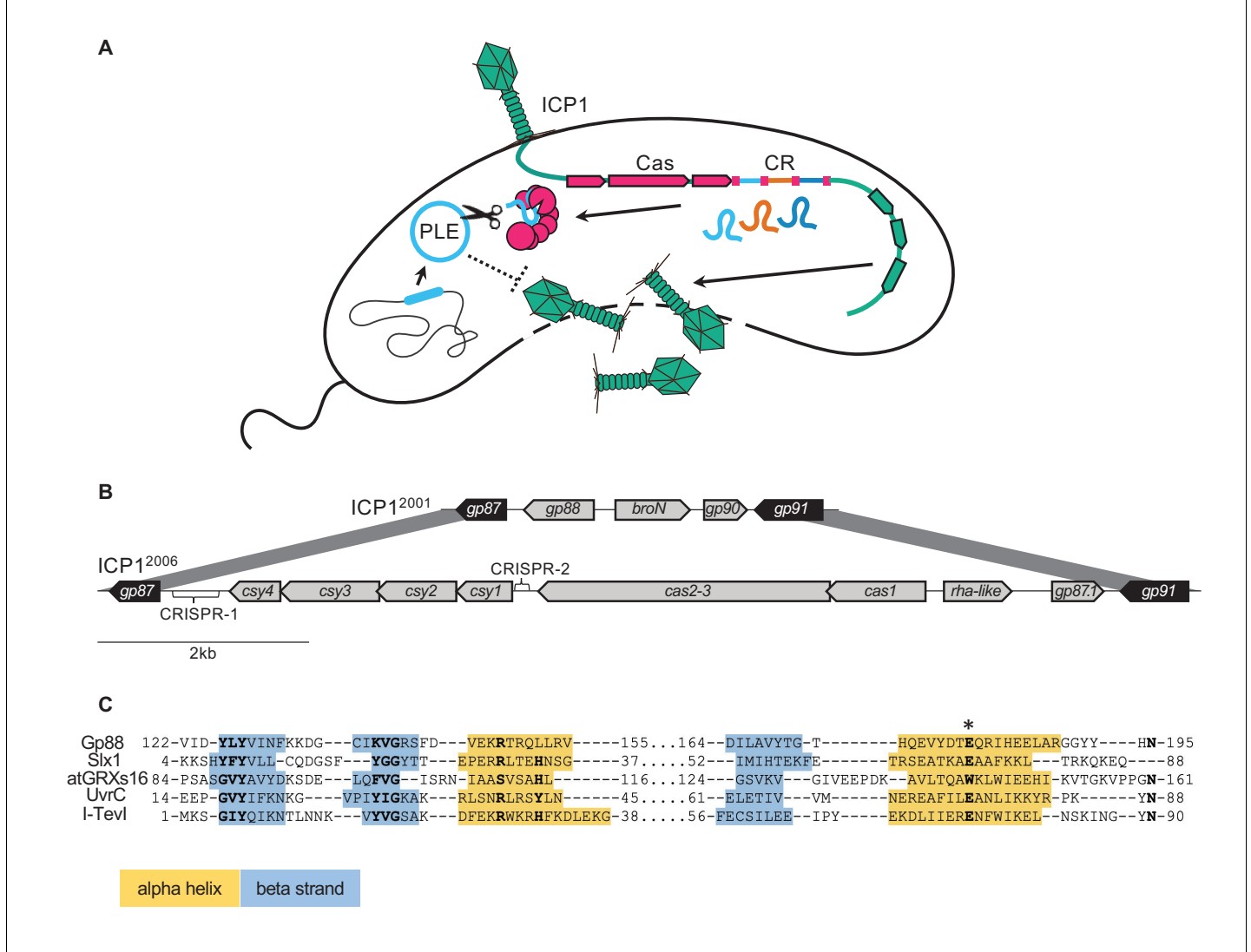

**Figure 1.** Some ICP1 isolates encode a free-standing nuclease in place of CRISPR-Cas. (**A**) A model of ICP1 interference of phage-inducible chromosomal island-*like* elements (PLEs) via CRISPR. When ICP1 infects a PLE(+) *V. cholerae* cell, ICP1 is able to overcome PLE restriction and reproduce if it possesses a CRISPR-Cas system with complementary spacers to the PLE. Cas and CR refer to the CRISPR associated genes and CRISPR array respectively. (**B**) Schematics of the region between *gp87* and *gp91* as it appears in ICP1[2001] (top) and ICP1[2006] (bottom). Genes represented by black arrows are conserved in all ICP1 isolates, while genes represented with gray arrows covary with *gp88* or CRISPR-Cas. (**C**) An alignment between the T5orf172 domain of Gp88 and the GIY-YIG domains of several structurally resolved endonucleases. Secondary structure for Gp88 was predicted using HHPRED (*Zimmermann et al., 2018*). Alpha helices are shown in yellow shading, and beta strands are shown in blue shading. Key residues of the GIY-YIG motif are bolded. We included an atypical GIY-YIG endonuclease domain from a chloroplast-encoded glutoredoxin atGRXs16 to demonstrate the potential for alternative residues at core motif positions. A conserved glutamate that was previously found to be required for catalysis in I-TevI is denoted by an asterisk (and corresponds to the E180A mutation in Gp88 in subsequent experiments).

The online version of this article includes the following figure supplement(s) for figure 1:

**Figure supplement 1.** ICP1[2005] has an atypical CRISPR-Cas arrangement.

isolates that lack CRISPR-Cas. Surprisingly, we found that all natural ICP1 isolates that do not encode CRISPR-Cas instead encode an endonuclease in the same genomic locus that is necessary for propagation on *V. cholerae* strains containing PLEs 1, 4, or 5. Lending further support to the 'guns for hire' model, we find that this anti-PLE nuclease is of chimeric origin, being partially derived from a PLE-encoded DNA-binding domain while its nucleolytic domain appears to be derived from an ICP1-encoded family of putative HEGs. Harnessing the rich evolutionary interplay of PLE and ICP1, this work shows that domain shuffling between hostile genomes can allow for new forms of

antagonism, and that phage-encoded HEGs can be repurposed for antiparasite functions. Additionally, this work reveals key mediators of ICP1-PLE host range that inform observed patterns of PLE temporal succession and modularity, broadening our understanding of subcellular host-parasite coevolution.

## Results

### A subset of ICP1 isolates deploy a stand-alone nuclease instead of CRISPR-Cas to counter PLE

We set out to identify which gene(s) determined host range in ICP1 isolates that lack CRISPR-Cas. It has long been recognized that phages are mosaic entities composed of functional gene neighborhoods, and syntenic neighborhoods of divergent sequence may fulfill analogous functions (*Brüssow and Hendrix, 2002*). Previous work suggests that ICP1 conforms to these general patterns of phage genome structure. Transcriptomics and bioinformatic predictions show that ICP1 genes with related biological functions are organized together in the genome and expressed at the same time, demonstrating the presence of gene neighborhoods (*Barth et al., 2020a*). Additionally, while the ICP1 genome is highly conserved between isolates and does not display large-scale rearrangements (*Angermeyer et al., 2018*), there is indication that nonhomologous sequence can serve analogous functions. ICP1 isolates encode one of two alternative SF1B-type helicases thought to be of shared function (*McKitterick et al., 2019a*), suggesting that ICP1 isolates can use alternative genes to fulfill the same adaptational requirement. We reasoned that such genome organization and mosaicism warranted a 'guilt by location' approach to investigating gene function, and that the locus syntenic to CRISPR-Cas in those ICP1 isolates that lack CRISPR might hold clues as to how they overcome PLEs.

In isolates without CRISPR-Cas, the locus is replaced with a single open reading frame, designated *gp88* for its location in ICP1[2001], the original sequenced ICP1 isolate (*Seed et al., 2011*; *Figure 1B*). The two coding sequences immediately upstream of the CRISPR-Cas system and oriented divergently from the system are also replaced in the phage with *gp88*. These genes generally covary with the presence of the CRISPR-Cas system or *gp88*. Most ICP1 CRISPR-Cas systems are adjacent to a phage regulatory protein Rha domain (pfam09669) encoding gene. A gene encoding a Bro-N domain (pfam02498) and a KilAC domain (pfam03374) occurs adjacent to *gp88*. Their positions and putative annotations suggest that these divergently transcribed genes may have a regulatory function. In one sequenced ICP1 isolate with a functional CRISPR-Cas system (*O'Hara et al., 2017*), the *bro_N* domain coding gene and its partner are found instead of the *rha*-like gene containing pair, suggesting that these pairs are redundant in function or not involved in CRISPR-Cas activity (*Figure 1—figure supplement 1*).

Further analysis of *gp88* revealed that it encodes a T5orf172 domain (pfam10544) containing protein. This domain is a member of the GIY-YIG endonuclease domain superfamily, suggesting that Gp88 may be a nuclease. We aligned the Gp88 T5orf172 domain with GIY-YIG domains from endonucleases that had been biochemically characterized and structurally resolved (*Liu et al., 2013*; *Swapna et al., 2005*; *Truglio et al., 2005*; *Van Roey et al., 2002*; *Figure 1C*). Unlike most identified GIY-YIG nucleases, Gp88 lacks a conserved histidine or tyrosine in the first alpha helix of the GIY-YIG domain. However, Gp88 retains conservation of the catalytic arginine and glutamate. The relatively high motif conservation found in Gp88 suggests that the protein possesses endonucleolytic activity. Given that nucleases are a particularly prominent class of proteins engaged in conflicts between hosts, mobile elements and viruses (*Koonin et al., 2020*), and *gp88*'s syntenic location to ICP1's CRISPR-Cas system, we hypothesized that Gp88 interferes with PLE activity to protect ICP1 isolates that naturally lack CRISPR-Cas.

To test our hypothesis, we generated ICP1[2001] mutants with either an in-frame deletion of *gp88* or harboring a single amino acid substitution (E180A) predicted to abolish Gp88's nucleolytic activity (*Figure 1C*). We also used ICP1[2006] and a ΔCRISPR derivative to serve as controls for host susceptibility. As expected, the PLE (-) *V. cholerae* strain was susceptible to all ICP1 variants, and the ICP1[2006] ΔCRISPR variant was restricted by all PLEs (*Figure 2A*). CRISPR(+) ICP1[2006] was able to propagate on all strains except the one containing PLE3 as ICP1[2006] does not have a matching spacer against PLE3. ICP1[2001] was able to propagate on PLEs 1, 4, and 5 (*Figure 2A*), but it was

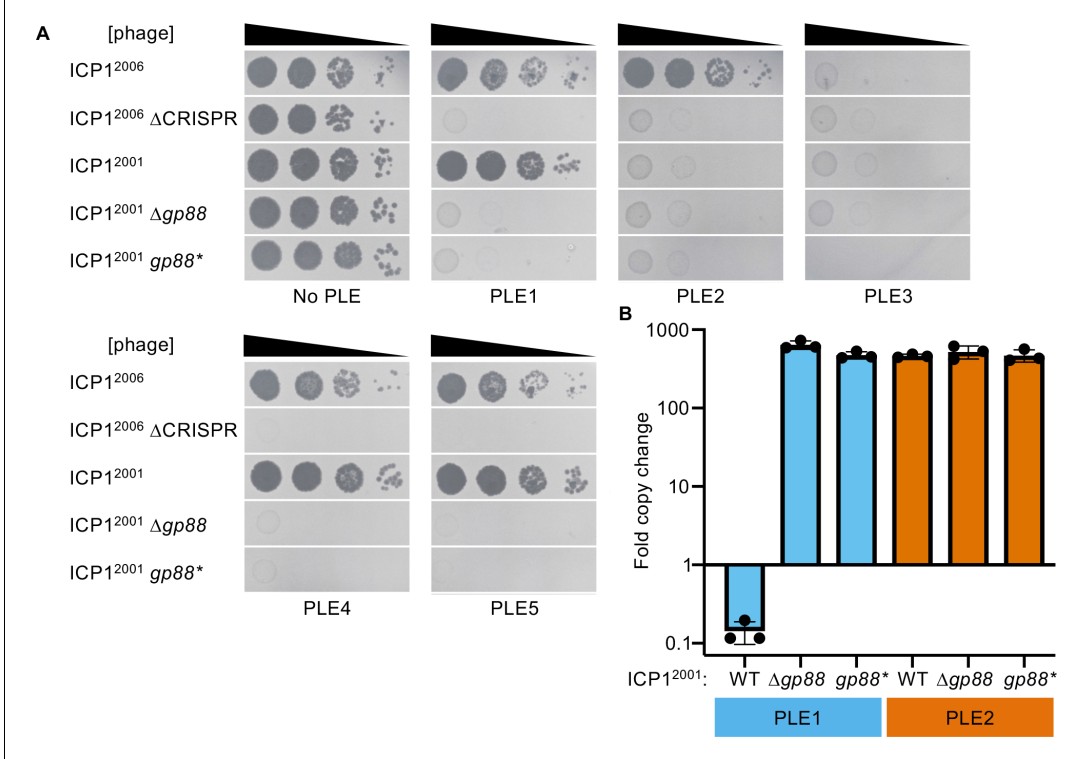

**Figure 2.** The alternative nuclease Gp88 controls ICP1 host range in a natural isolate that lacks CRISPR-Cas. (**A**) Tenfold dilutions of the phage isolate or mutant derivative indicated spotted on *V. cholerae* with the PLE indicated (bacterial lawns in gray, zones of killing are shown in black). Gp88* possess a single amino acid substitution (E180A) predicted to abolish nuclease activity. Spot assays were performed in biological triplicate, and a single representative image is shown. Replicate spot assays are shown in *Figure 2—figure supplement 1* and *Figure 2—figure supplement 2*. (**B**) Replication of PLE1 and PLE2 in *V. cholerae* host strains calculated as the fold change in PLE DNA copy 20 minutes post infection with the ICP1 variant indicated.

The online version of this article includes the following source data and figure supplement(s) for figure 2:

**Source data 1.** Values for the graph in *Figure 2B*.
**Figure supplement 1.** Biological replicate of spot assays.
**Figure supplement 2.** Biological replicate of spot assays.

restricted on PLEs 2 and 3. Unlike the wild-type (WT) variant, the Δ*gp88* and *gp88*\* mutants were unable to propagate on *V. cholerae* with PLEs 1, 4, or 5 (*Figure 2A*), indicating that catalytically active *gp88* is necessary for overcoming these PLEs in phages naturally lacking CRISPR-Cas. Unsurprisingly, the *gp88* mutants retained sensitivity to restriction by PLEs 2 and 3 (*Figure 2A*).

As CRISPR targeting was previously observed to diminish PLE replication that occurs during ICP1 infection (*McKitterick et al., 2019b*), we tested whether the presence of Gp88 could impact PLE replication during infection. We observed that PLE1 is unable to replicate in the presence of Gp88 encoding ICP1, and replication is restored during infection with the Gp88 knockout or catalytically inactive mutant phages (*Figure 2B*). Consistent with endonucleolytic activity, the PLE1 copy decreases following infection by Gp88 encoding phage. This is notable given that previous work has shown that multiple PLE matched spacers are required for CRISPR-Cas to completely abolish PLE replication during infection (*McKitterick et al., 2019b*). PLE2 replication is unaffected by the presence or absence of Gp88, consistent with Gp88 not providing ICP1 with immunity against PLE2 (*Figure 2A*).

## PLE replicons are modular

Having identified *gp88*'s role in preventing PLE restriction of ICP1, we next sought to determine how Gp88 was recognizing PLE, and why it did not confer protection against PLEs 2 and 3. We reasoned that PLEs 2 and 3 most likely lacked sequence targeted by Gp88 or encoded an inhibitor of

Gp88's activity. To explore these possibilities, we compared the PLE genomes looking for nucleotide sequence that was uniquely present or uniquely absent in PLEs 2 and 3.

Strikingly, only two stretches of sequence met these criteria, and both had been previously implicated in PLE replication (Barth et al., 2020b). The repA gene encoding the replication initiation factor, and the intergenic region containing the PLE origin of replication (ori) to which RepA binds covaried, with the PLE1, 4, and 5 sequences clustering together as one group, and the PLE2 and 3 sequences clustering as another (Figure 3A). More specifically, it was the DNA-binding RepA_N

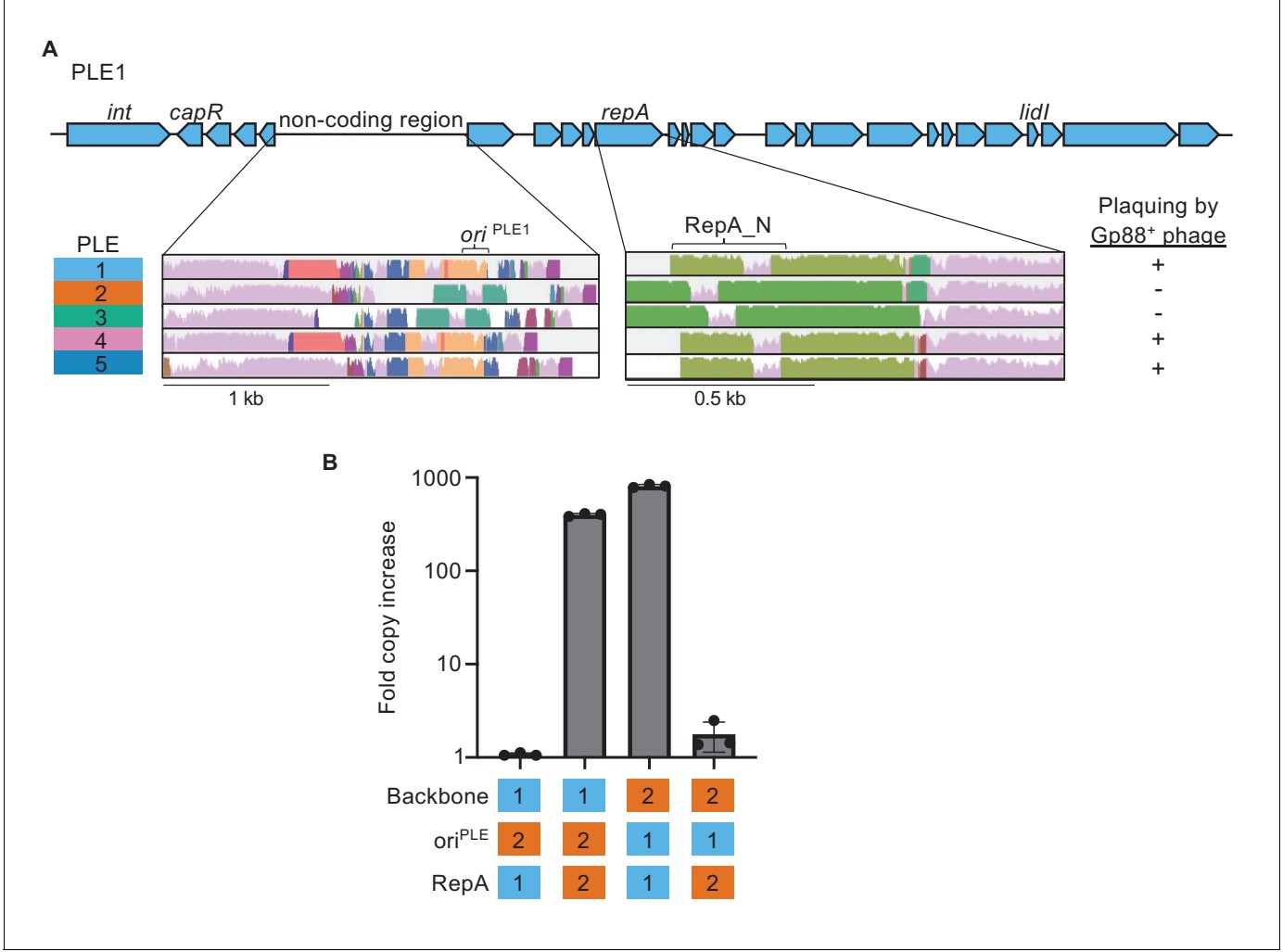

Figure 3. Phage-inducible chromosomal island-like element (PLE) replicons are modular and are composed of a compatible RepA initiation factor and origin of replication (ori). (A) Genomic organization of PLE1 with insets corresponding to the PLE noncoding region and repA. Previously characterized PLE genes are labeled. Insets are Mauve alignments showing sequence conservation of the denoted loci across the different PLEs. Shared color denotes sequence conservation, with the height of the histogram representing nucleotide sequence identity. The susceptibility of each PLE(+) V. cholerae host to plaquing by phage encoding Gp88 is indicated. (B) Replication of hybrid PLEs in V. cholerae calculated as the fold change in PLE DNA copy 20 minutes post infection with ICP1²⁰⁰⁶ ΔCRISPR. Strains with PLE1 ΔrepA (possessing the native ori^PLE1 or Δori::ori^PLE2) or PLE2 ΔrepA (possessing the native ori^PLE2 or Δori::ori^PLE1) were complemented with a vector expressing the repA gene from PLE1 or PLE2. The backbone, identity of the ori, and RepA variant are indicated as being from PLE1 or PLE2.

The online version of this article includes the following source data and figure supplement(s) for figure 3:

Source data 1. Values for the graph in Figure 3B.
Figure supplement 1. Alignment of repA from phaPLEs 1–5 encoding the conserved C-terminus.
Figure supplement 2. PLE2 requires RepA and a cognate origin of replication to replicate.
Figure supplement 2—source data 1. Values for the graph in Figure 3—figure supplement 2.

domain of RepA that covaried with the origin, while the C-terminal domain, hypothesized to facilitate replisome recruitment (*Barth et al., 2020b*), was conserved across all PLEs (*Figure 3A*).

Previously, we found that during ICP1 infection, ectopic expression of RepA was sufficient to drive replication of a synthetic 'midiPLE' construct. The midiPLE consists of the PLE attachment sites, the PLE integrase, and the noncoding region that bears the origin of replication (*Barth et al., 2020b*). Additionally, midiPLE replication did not occur without RepA, and the PLE integrase was shown to be dispensable for PLE replication (*Barth et al., 2020b*). These data suggest that the minimal components of the PLE replicon are the replication origin and RepA, the two components that covaried across PLEs. Alignment of the conserved 3′ ends of PLE *repA* genes suggests that RepA specificity swapping has occurred multiple times (*Figure 3A*, *Figure 3—figure supplement 1*). Despite the PLE1 RepA_N domain clustering with the PLE4 and PLE5 variants, the PLE1 C-terminal sequence is more similar to PLE2 (98.88% identical over the last 178 bp) than PLE5 (93.25% identical over the last 178 bp). PLE5 and PLE3 are 99.44% identical over the same region, while the PLE4 C-terminal region is the most diverged from other PLEs (*Figure 3—figure supplement 1*). This cross-clustering of *repA* ends would only be expected to occur after multiple gene recombination events, suggesting that PLE replisome module swapping occurred at least twice, and may be an important part of PLE evolution.

The putative modularity of the PLE origins and RepA_N domains covaried with susceptibility to ICP1[2001] (*Figure 3A*), leading us to hypothesize that one of the replicon modules but not the other was susceptible to Gp88-mediated interference. Before testing this hypothesis directly, we wanted to confirm that the covariation of PLE replication origins and RepA_N domains truly reflects modularity of the PLE replicon. We first tested whether the putative PLE2 origin and *repA* gene were necessary for PLE2 replication and found that deletion of either component abolished replication following infection by ICP1[2006] ΔCRISPR (*Figure 3—figure supplement 2*), as was previously observed for PLE1 (*Barth et al., 2020b*).

Having confirmed that the PLE2 variant replicon components are necessary for replication, we then sought to demonstrate specificity of the RepA variants to their cognate origin of replication. We generated chimeric 'origin-swapped' Δ*repA* PLEs for PLEs 1 and 2 (*Figure 3B*), and ectopically expressed each RepA variant in the different PLE backgrounds during phage infection. As expected, PLE replication only occurred when cognate origins and *repA* alleles co-occurred, revealing that the two components of the PLE replicon function together as a module, irrespective of which PLE backbone they are encoded in.

## Gp88 is an origin-directed nuclease

Having established the specificity between RepA variants and their cognate origins of replication and recognizing that sensitivity to Gp88 covaried with replicon type, we took a closer look at Gp88 to decipher how it might interface with the PLE replication module. Remarkably, Gp88's own N-terminal domain is 42% identical and 61% sequence similar across 93% of PLE1's RepA_N domain (*Figure 4A*). This was surprising as Gp88's T5orf172 domain is similar to those of several putative HEGs within the ICP1 genome (*Figure 4—figure supplement 1*). The high similarity of Gp88's N-terminal portion to some PLE-encoded RepA alleles and the C-terminal portions similarity to putative HEGs suggest that *gp88* may have arisen as a chimeric hybrid of PLE and ICP1 coding sequences. Additionally, the similarity of Gp88's N-terminal region to PLEs 1, 4, and 5 RepA DNA-binding domains suggested that Gp88 might bind to the replication origins of PLEs 1, 4, and 5 and cleave at or proximal to that site.

To evaluate this hypothesis, we next wanted to test whether the PLE origin of replication was a necessary component for Gp88 activity. Previously, it was shown that loss of replication partially attenuated PLE1-mediated restriction of ICP1 but nonreplicating PLE1 mutants were still broadly restrictive to ICP1 ΔCRISPR-Cas (*Barth et al., 2020b*). We hypothesized that PLE could escape Gp88 targeting through deletion of the PLE origin and thus block propagation of Gp88 encoding phage. We tested this by deleting the entire conserved stretch of sequence that contained the origin of replication in PLEs 1, 4, and 5. In support of our hypothesis, these mutants regained restrictive activity against Gp88 encoding phage (*Figure 4B*). Conversely, cloning a Gp88 recognized replication origin into PLEs that are insensitive to Gp88 should sensitize them to Gp88 activity. To test this, we infected our 'ori-swapped' PLE2 strain (*Figure 3B*) with Gp88 encoding phage. We included a PLE2 Δ*ori* strain to control for the possibility that loss of replication would abolish PLE2-mediated

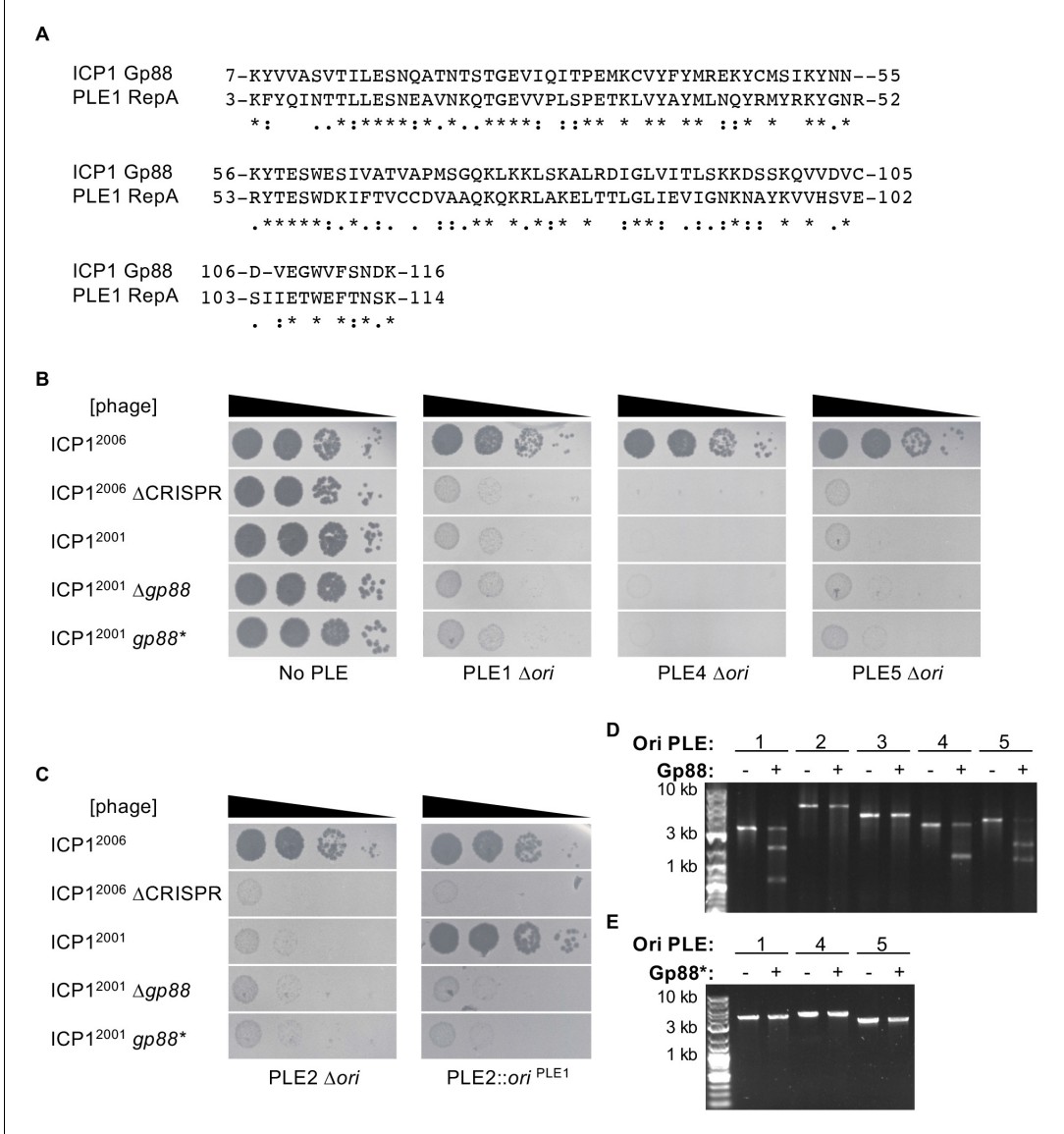

**Figure 4.** Gp88 is a PLE replication origin-directed nuclease. (**A**) Sequence alignment of the N-terminal portion of Gp88 with the RepA_N domain from PLE1 RepA. Identical residues are denoted with a '*.' Strong residue similarity is denoted by ':', and weak similarity is denoted by '•.' (**B, C**) Tenfold dilutions of the phage isolate or mutant derivative indicated spotted on *V. cholerae* with the PLE indicated (bacterial lawns in gray, zones of killing are shown in black). Spot assays were performed in parallel with those in *Figure 2*, and images labeled with the same PLE background are the same image. Spot assays were performed in biological triplicate, and a single representative image is shown. Biological replicates are shown in *Figure 2—figure supplement 1*. Gp88* possess a single amino acid substitution (E180A) predicted to abolish nuclease activity. (**B**) shows phage susceptibility of *V. cholerae* with PLE1, PLE4, and PLE5 Δ*ori* derivatives as compared to a strain without PLE. (**C**) shows phage susceptibility for *V. cholerae* with PLE2 Δ*ori* and PLE2 Δ*ori::ori*^PLE1^. (**D**) Nuclease assay showing the integrity of a PCR product amplified from the noncoding region containing the ori from the PLE variant indicated (numbers) treated with (+) and without (–) 500 nM of purified Gp88. Nuclease assays were performed in triplicate, replicates are presented in *Figure 4—figure supplement 3*. (**E**) Nuclease assay showing the integrity of a PCR product amplified from the noncoding region containing the ori from the PLE variant indicated (numbers) treated with (+) and without (–) 500 nM of purified Gp88*. Nuclease assays were performed in triplicate, replicates are presented in *Figure 4—figure supplement 4*.

The online version of this article includes the following source data and figure supplement(s) for figure 4:

**Source data 1.** Original uncropped gels for *Figure 4D* (top) and *Figure 4E* (bottom).

**Figure supplement 1.** The nuclease domain of Gp88 is similar to those of putative homing endonucleases in ICP1.

**Figure supplement 2.** Protein preparations of Gp88 (Odn) and Gp88* (Odn*) used for in vitro assays.

**Figure supplement 2—source data 1.** Original uncropped gels for *Figure 4—figure supplement 2*.

**Figure supplement 3.** Replicates of Gp88 nuclease assay.

**Figure supplement 3—source data 1.** Original uncropped gels for *Figure 4—figure supplement 3*.

*Figure 4 continued on next page*

**Figure supplement 5.** Cleavage of an altered PLE4 probe.

restriction of ICP1. The PLE2 Δ*ori* strain retained the ability to restrict all isolates of ICP1$^{2001}$ (*Figure 4C*). In contrast, the PLE2 strain bearing the PLE1 origin sequence was no longer restrictive to ICP1$^{2001}$, but still restricted variants that lacked Gp88 activity (*Figure 4C*), confirming that the presence of the replication origin sequence mediated sensitivity to Gp88.

We wanted to confirm that Gp88-mediated interference manifested through nucleolytic cleavage of PLE. To determine if Gp88 was truly acting as a nuclease, we purified Gp88 (*Figure 4—figure supplement 2*) and performed in vitro nuclease activity assays. Consistent with the host range of ICP1 encoding Gp88, we found that the purified Gp88 protein cut PCR products amplified from the region containing the origin of replication from PLEs 1, 4, and 5, but did not cut those of PLEs 2 and 3 (*Figure 4D*), confirming that Gp88 disrupts PLE through nuclease activity. Supporting this interpretation, the E180A Gp88* mutant that was inactive against PLEs 1, 4, and 5 in vivo, and predicted to be catalytically inert, did not cleave PCR products amplified from PLEs 1, 4, and 5 (*Figure 4E*), further linking the in vivo activity of Gp88 to its capacity to cleave in vitro. Curiously, Gp88 produced only one cleavage product from the PLE4 probe. We reasoned that since the PLE noncoding regions are diverse and the PLE4 origin of replication was located near to the center of the PLE4 probe, Gp88 cleavage of the PLE4 probe might produce two bands of indistinguishable size. To check this, we produced a new PLE4 probe with the origin of replication offset from the middle and found that Gp88 produced two visible bands (*Figure 4—figure supplement 5*). In light of these results, we renamed Gp88 the origin-directed nuclease or Odn.

## Odn requires iterons to cleave the PLE origin of replication

Our results so far suggested a model where Odn mimics the specificity of the PLE1, 4, and 5 RepA proteins to bind and cut at their cognate origins of replication. Previously, PLE1 RepA was found to bind specifically to a set of iterons, a series of three ~30 bp semi-palindromic repeats in the PLE1 origin of replication (*Barth et al., 2020b*). If Odn specificity truly mimicked that of RepA, then it should require the iteron sequence for cutting. We tested this in vitro by titrating increasing concentrations of Odn in a nuclease assay with the WT PLE1 origin of replication, as well as the same substrate except with the iterons deleted. In support of our model, we found that Odn does require the iteron sequence for cleavage (*Figure 5A*). Consistent with iterons being necessary for Odn-mediated in vitro cleavage of the PLE origin, ICP1$^{2001}$ infection was restricted by a PLE1 strain harboring the same iteron deletion (*Figure 5B*). Together, these results strongly support that Odn has DNA-binding specificity that mimics that of the replication initiation factor of some PLEs.

## PLE mutations lead to escape from Odn

Odn activity against the origin, as well as the pattern of cross-clustering at the N and C termini of RepA, suggested that Odn may impose substantial evolutionary pressure on PLE replication modules. Since swapping the PLE origin and cognate RepA_N domain could abolish Odn targeting of PLE, it appears likely that Odn selected for the multiple domain shuffling events in PLE RepA inferred by comparison of PLE genomes (*Figure 3—figure supplement 1*). Because PLE replication is necessary for both PLE mobility and complete restriction of ICP1 (*Barth et al., 2020b*; *McKitterick et al., 2019a*), simple deletions of the replication origin would not likely be favored as long-term solutions to evading recognition and subsequent cleavage by Odn.

While the swapping and diversification of certain sequences can be traced through the five PLEs, each PLE variant is remarkably conserved. All members of each variant have been found to be 100% nucleotide identical in previously published data sets (*McKitterick et al., 2019b*; *O'Hara et al., 2017*). However, we found a single instance of diversity in PLE1 within a lineage of *V. cholerae* isolated from Pakistan. This lineage was represented in five sequenced strains (biosample accession numbers SAMN08979118, SAMN08979175, SAMN08979185, SAMN08979188, and

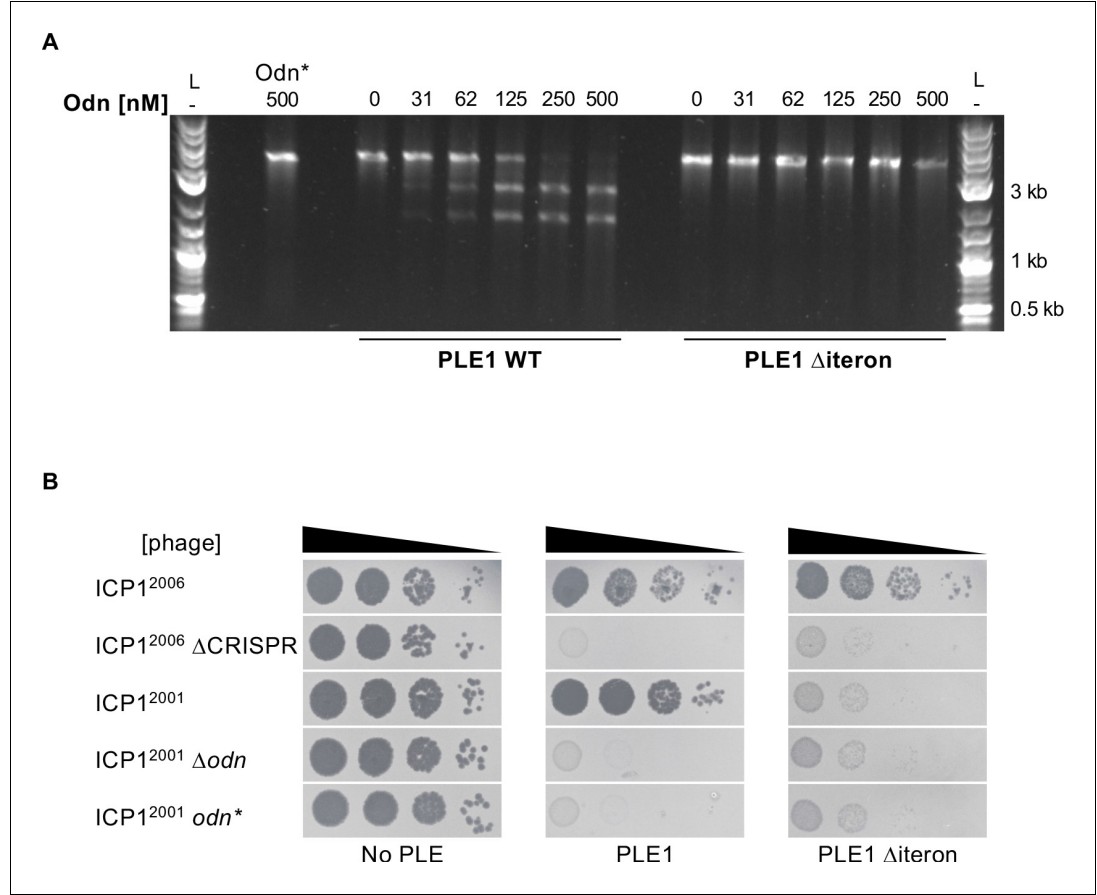

**Figure 5.** ICP1-encoded Odn (Gp88) requires the PLE iterons for cleavage. (**A**) Nuclease assay showing the integrity of a PCR product amplified from the noncoding region containing the ori from wild-type (WT) PLE1 and the Δ*iteron* mutant, with purified Odn (31.25–500 nM) titrated in. 500 nM catalytically inactive Odn (Odn*) with a single amino acid substitution (E180A) with the WT PLE1 sequence was also included (far left). Nuclease assays were performed in triplicate and replicates are presented in *Figure 5—figure supplement 1*. (**B**) Tenfold dilutions of the phage isolate or mutant derivative indicated spotted on *V. cholerae* with the PLE indicated (bacterial lawns in gray, zones of killing are shown in black). Spot assays were performed in parallel with those in *Figures 2* and *4*, and images labeled with the same PLE background are the same image. Spot assays were performed in biological triplicate, and a single representative image is shown. Replicate assays are shown in *Figure 2—figure supplement 1*.

The online version of this article includes the following source data and figure supplement(s) for figure 5:

**Source data 1.** Original uncropped gel for *Figure 5A*.

**Figure supplement 1.** Replicates of nuclease assays.

**Figure supplement 1—source data 1.** Original uncropped gels for *Figure 5—figure supplement 1*.

---

SAMN08979253). In these five strains, we discovered variation in a 67 bp stretch covering the iterons that results in several nucleotide changes. Within the first iteron, there is an A to T transversion, and starting at that transversion, the next 42 bp are duplicated and replace the sequence that is normally downstream (*Figure 6A*). This change maintains the presence of the three iterons, and even reverts a few variant bases to ones in the PLE4 and 5 iterons (*Figure 6B*). Notably, these changes are the only sequence differences between these PLE variants and all other PLE1 isolates, aside from a 2 bp extension of an 11 bp polyA tract that also occurs in these atypical PLE1 variants.

We sought to test if this natural example of PLE1 iteron diversity had any effect on susceptibility to Odn. We generated a DNA probe covering this variant region (denoted PLE1^Mut) and tested its sensitivity to Odn in comparison to the WT PLE1 ori sequence. The PLE1^Mut sequence was notably less susceptible to cleavage by Odn than the WT allele, but some cutting of the PLE1^Mut probe at the highest concentration of Odn was apparent (*Figure 6C*). This raised the question of whether this

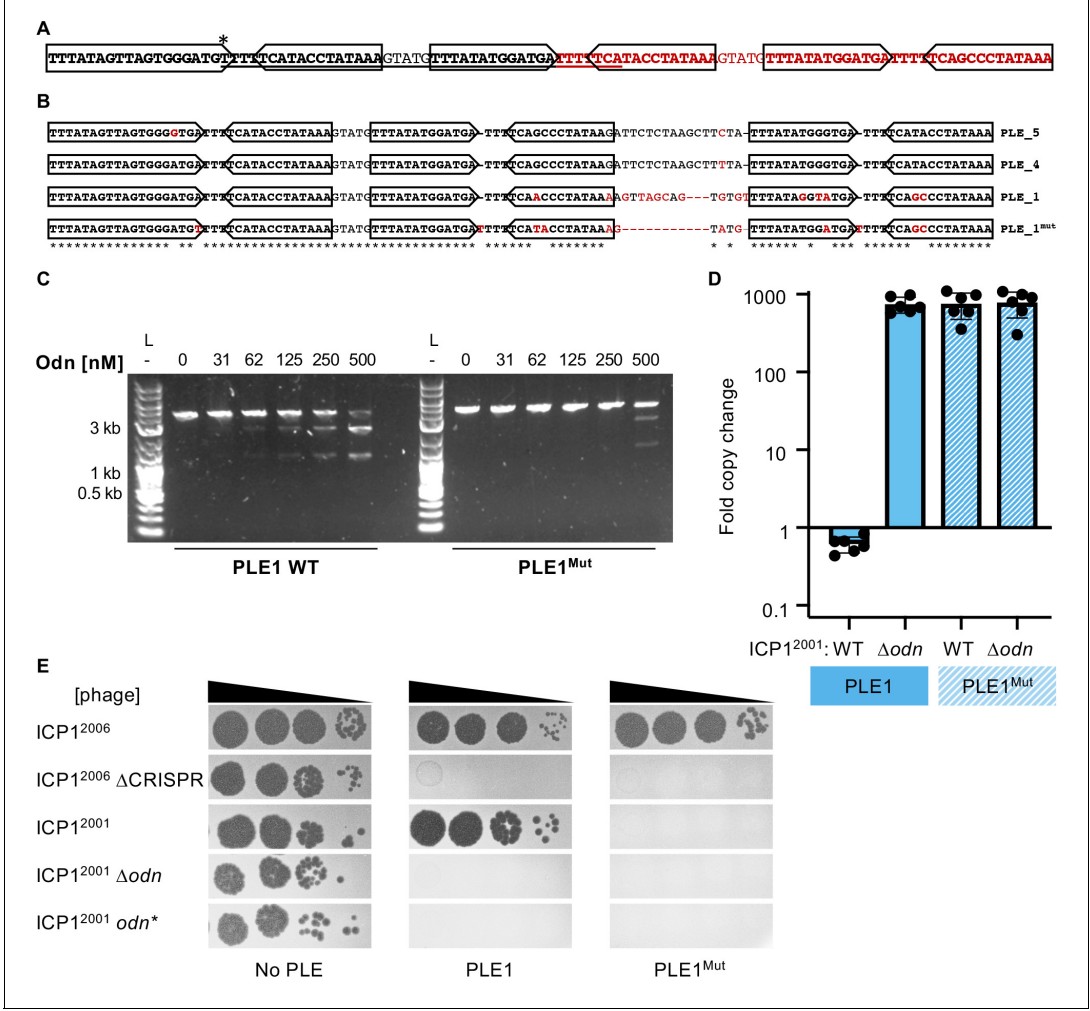

**Figure 6.** Mutations in PLE1 present in *V. cholerae* isolates from Pakistan render the PLE1 ori resistant to Odn-mediated cleavage. (**A**) The PLE1$^{Mut}$ iteron region. Iterons are bolded and sub-repeats are denoted with arrows. The underlined sequence is identical to the sequence in red. An asterisk (*) denotes the location of an A to T substitution. (**B**) An alignment of the iterons from PLE5, PLE4, PLE1, and PLE1$^{Mut}$. Iterons are in bold with sub-repeats indicated with arrows. Sequence deviating from a consensus is shown in red. Regions with 100% conservation are denoted with an asterisk. (**C**) Nuclease assay showing the integrity of a PCR product amplified from the noncoding region containing the ori from wild-type (WT) PLE1 or PLE1$^{Mut}$ with purified Odn (31.25–500 nM) titrated in. Nuclease assays were performed in triplicate, and replicates are presented in *Figure 6—figure supplement 1*. (**D**) Replication of PLE1 WT and PLE1$^{Mut}$ in *V. cholerae* calculated as the fold change in PLE DNA copy 20 minutes post infection with the ICP1 variant indicated. (**E**) Tenfold dilutions of the phage isolate or mutant derivative indicated spotted on *V. cholerae* with the PLE indicated (bacterial lawns in gray, zones of killing are shown in black). Biological replicates of the spot assays are presented in *Figure 6—figure supplement 2*. The online version of this article includes the following source data and figure supplement(s) for figure 6:

Source data 1. Original uncropped gel for *Figure 6C*.
Source data 2. Values for the graph in *Figure 6D*.
Figure supplement 1. Replicates of nuclease assays.
Figure supplement 1—source data 1. Original uncropped gels for *Figure 6—figure supplement 1*.
Figure supplement 2. Biological replicates of PLE1$^{Mut}$ spot assays.

PLE1 variant could resist attack by Odn in vivo. To investigate this possibility, we cloned the PLE1$^{Mut}$ origin of replication into our PLE1 strain and challenged this mutant with WT, Δ*odn* and *odn*\* ICP1$^{2001}$. The PLE1$^{Mut}$ origin of replication was able to drive replication to levels comparable to that of the WT origin, and robust replication was maintained even in the presence of Odn (*Figure 6D*). In agreement with our replication data, we also found that our PLE1$^{Mut}$ strain restricted production of WT ICP1$^{2001}$ (*Figure 6E*). These data demonstrate that PLE can escape Odn activity through subtle

restructuring of the iterons, in addition to more extensive replication module exchange (*Figure 3A*), and suggest that ICP1 counter-defenses like Odn may select for diversification of the PLE replication machinery.

## Discussion

Here, we have identified a new anti-PLE effector present in some ICP1 isolates and identified its mode of anti-PLE activity. The anti-PLE effector Odn possess two domains, an endonuclease domain and a DNA-binding domain, which has sequence similarity to the DNA-binding domain of the RepA initiation factor from PLEs 1, 4, and 5. Mimicking RepA's specificity, Odn cuts the origin of replication in those PLEs. PLEs are able to escape Odn antagonism through mutation of their origin of replication, and there is compelling evidence suggesting that PLEs have exchanged replicon modules for an alternative replication origin and initiation factor origin-binding domain on at least two separate occasions. While sampling of ICP1 isolates and *V. cholerae* strains is far from complete, we do see the oldest ICP1 strains encode Odn and the oldest PLE variants encode the sensitive origin of replication. It is possible that Odn selected for alternative replicon modules, leading to the decline of PLE4 and succession of PLE2 that occurred in the early 2000s (*O'Hara et al., 2017*). This may in turn have caused selection for the presence of CRISPR-Cas, a decrease in prominence of Odn, and the reemergence of the sensitive replicon module in PLE1 matching the trends of PLE and ICP1 occurrence observed thus far (*McKitterick et al., 2019b*; *O'Hara et al., 2017*). This model of PLE and ICP1 succession is consistent with an antagonistic frequency-dependent selection (aFDS) mechanism of host-pathogen coevolution where some level of genetic diversity is maintained among antagonistic genes and rare alleles are selected for, leading to oscillation of the dominant genotype (*Papkou et al., 2019*). In some experimental evolution studies using bacteriophages, aFDS dynamics are found to eventually develop (*Gómez and Buckling, 2011*; *Hall et al., 2011*; *Lopez Pascua et al., 2014*). This seems a likely possibility for PLEs and ICP1 whose specific adaptations and counter adaptations suggest that the two genomes have been coevolving for a long period of time.

It is interesting to consider why a specialized anti-PLE mechanism might persist within some ICP1 isolates when adaptive CRISPR-Cas immunity exists as an alternative. An obvious benefit of Odn is its small size; *odn* is less than 700 bp and more than 10 times smaller than ICP1's CRISPR-Cas system. Given that ICP1's genome size is limited by what it can package into its capsid, the extra sequence taken up by CRISPR-Cas could be better spent on other auxiliary genes if it is not needed to overcome PLE. Additionally, Odn may provide more complete restriction of certain PLEs. While a single CRISPR spacer is sufficient to enable ICP1 plaque formation on PLE(+) cells, it was previously found that multiple spacers were needed to completely abolish PLE replication, and PLE transduction could still be detected from single-spacer ICP1 infections (*McKitterick et al., 2019b*; *O'Hara et al., 2017*). Additionally, different CRISPR spacers had different outcomes in terms of PLE transduction and ICP1 reproduction (*McKitterick et al., 2019b*), suggesting that where PLE is targeted may be important. Position-dependent outcomes for cleavage may explain why Odn is sufficient to provide robust interference against the PLEs it targets. By destroying the origin of replication, Odn-mediated PLE degradation might not have to compete with PLE replication and could head off pathways for PLE escape through recombinational repair. It should also be considered that certain environmental and physiological conditions as well as some genetic factors might render ICP1's CRISPR-Cas system less effective. While no PLEs appear to encode anti-CRISPRs, numerous anti-CRISPRs have been found in Gammaproteobacteria (*Pawluk et al., 2016*), and co-occurrence of these genes with PLE could neutralize ICP1's CRISPR-Cas as an anti-PLE strategy. Overall, it appears that Odn lacks the flexibility provided by CRISPR adaptation but provides more reliable interference against a subset of PLE variants.

In addition to ICP1 and Odn, at least one other anti-PLE mechanism has evolved as some CRISPR (+) ICP1 isolates are able to plaque on PLE2 when CRISPR is deleted (*O'Hara et al., 2017*). Whatever this anti-PLE mechanism may be, it may have helped select against PLE2, leading to the emergence of PLE1 as another possible example of an aFDS dynamic in the ICP1-PLE arms race. It seems unlikely that this anti-PLE2 mechanism is specific to the alternative replicon module as PLE3, which has a similar replicon to PLE2, maintains restriction of those particular ICP1 isolates (*O'Hara et al., 2017*). In any case, the co-occurrence of a separate anti-PLE mechanism in the same phage as a

CRISPR-Cas system further suggests that CRISPR-Cas may have weak spots in terms of PLE inhibition.

It is somewhat surprising that the PLE1$^{Mut}$ variant is able to escape Odn activity, given that the new iteron sequences are largely similar to those that exist in PLEs 1, 4, and 5. The most notable differences are an extra T/A base between the inverted sub-repeats in both the second and third iterons, and a reduction of space between the second and third iterons (now 5 bp instead of 14 or 16 bp) (*Figure 6B*). These changes appear more consistent with some sort of steric effect on catalysis from improper spacing rather than a loss of sequence recognition; however, the molecular details of Odn binding and catalysis remain to be elucidated. The T4-encoded GIY-YIG HEG I-TevI is known to have two separate target specificities, one for DNA binding and one for cleavage, and cleavage is only efficient if the two recognition sites are properly spaced (*Liu et al., 2006*). It is possible that Odn nucleolytic activity also has some sequence specificity, and closer spacing of the iterons is sufficient to block binding and cleavage by Odn. Based on the data presented here, it is not possible to infer if Odn nucleolytic activity has sequence specificity, and what that specificity might be. For I-TevI, the sequence requirements for cleavage are somewhat loose (*Roy et al., 2016*), making cleavage sites harder to predict from sequence alone.

Alternatively, Odn cleavage might require recruitment of multiple Odn proteins, spaced a certain distance apart. RepA is thought to bind as dimer (*Barth et al., 2020b*), but at least some GIY-YIG endonucleases, including I-TevI, function as monomers (*Van Roey et al., 2002*). It is conceivable that Odn could require cooperative binding for catalysis or alternatively bind and cut at multiple subrepeats independently. While our results show robust double-stranded cleavage by Odn, this could be achieved through multiple single-strand nicks from appropriately spaced iterons. Odn and RepA provide an interesting example of proteins with overlapping sequence specificity. Working out the intricacies of their binding specificity and activity may prove fruitful for understanding how DNA-binding proteins evolve new targets and functions.

One of the most compelling aspects of Odn is its evolutionary relationship to a family of putative HEGs. HEGs are usually considered selfish genetic elements and are common in phage genomes (*Edgell, 2009*; *Edgell et al., 2010*; *Stoddard, 2011*). This adds a layer of symmetry to the *V. cholerae*-ICP1-PLE arms race. To defend itself against ICP1, *V. cholerae* makes use of the PLE, a selfish MGE. To protect against PLE, ICP1 has repurposed a nuclease domain from its own HEG parasites. While this recruitment of MGEs for antagonistic functions fits nicely within the 'guns for hire' model of MGE evolution, we see an interesting twist where it seems the horizontal gene transfer of DNA-binding domains is a central mediator of the PLE-ICP1 conflict. Recently, it was found that PLEs reduce ICP1 capsid gene expression through use of a regulator that resembles the DNA-binding domains of ICP1-encoded HEGs (*Netter et al., 2021*), the very same family of genes from which Odn's nuclease domain is derived (*Figure 4—figure supplement 1*). While viral 'capture' of host genes by genetic parasites for the purpose of manipulating gene expression is well described in both phage and eukaryotic viruses (*Alcami, 2003*; *Bryan et al., 2008*; *Zeng and Chisholm, 2012*), to our knowledge Odn is the first example of an anti-MGE gene where a sequence-specific DNA-binding protein has been 'captured' and fused to an endonuclease domain as a means of destroying the very genome that the sequence originated from. This highlights how horizontal gene transfer can serve as a shortcut to acquiring new sequence-specific antagonists during antagonistic coevolution and provides a slower evolving parallel to adaptive immunity through spacer acquisition.

It is striking that two examples of horizontal transfer between ICP1 and PLE relate to the same family of HEGs. Domain shuffling has been described for HEGs previously (*Landthaler and Shub, 2003*) and is likely especially adaptive for HEGs, which are thought to amplify within host genomes by acquiring new sequence specificities (*Gogarten and Hilario, 2006*; *Roy et al., 2016*). The domain architecture of some phage-encoded HEGs has been likened to 'beads on a string': with independent functional domains connected by linker sequences (*Van Roey and Derbyshire, 2005*). It is conceivable that proteins with this architecture might be more amenable to domain shuffling, and given the selfish nature of HEGs, they may have adaptations to tolerate or promote this shuffling as a means of diversification and dispersal. While HEG domestication has been much discussed (*Coughlan et al., 2020*; *Stoddard, 2011*) and the connection between genome antagonism and nucleases is well established in bacteria (*Koonin et al., 2020*), a specific link between HEGs and MGE antagonism had not been established prior to characterization of Odn. It is not clear if Odn has any intrinsic homing activity, but Odn may be able to cleave and replace CRISPR-Cas loci that

have acquired spacers matching the Odn recognition sequence. This could layer a Odn vs. CRISPR-Cas genetic conflict on top of the ICP1-PLE arms race. Because of their selfish nature, ability to mobilize, nucleolytic activity, and enrichment within viral and mobile genomes, HEGs are poised to be at the forefront of antagonistic coevolution between viruses and other MGEs, and we anticipate that other putative HEGs have unrecognized antiparasite activities.

# Materials and methods

## Key resources table

| Reagent type (species) or resource | Designation | Source or reference | Identifiers | Additional information |
|---|---|---|---|---|
| Gene (Vibrio cholerae) | RepA$^{PLE1}$ (PLE1 ORF11) | *Barth et al., 2020b* | WP_002040284.1 | |
| Gene (Vibrio cholerae) | RepA$^{PLE2}$ (PLE2 ORF14) | This paper | AGG36643.1 | |
| Gene (Bacteriophage ICP1) | Odn (ICP1_2001_Dha_0 *gp88*) | This paper | YP_004251029 | |
| Gene (Bacteriophage ICP1) | Odn* (ICP1_2001_Dha_0 *gp88$^{E180A}$*) | This paper | | The E180A mutation is predicted to abolish catalytic activity |
| Recombinant DNA reagent | P$_{tac}$-*repA*$^{PLE1}$ (plasmid) | *Barth et al., 2020b* | pZKB129 | Inducible RepA from PLE1 |
| Recombinant DNA reagent | P$_{tac}$-*repA*$^{PLE2}$ (plasmid) | This paper | pKS2159 | Inducible RepA from PLE2 |
| Recombinant DNA reagent | pE-SUMO-Odn (plasmid) | This paper | pKS2187 | Vector to express 6xHisSumo-fusion protein, fused to N-terminus of Odn (Gp88) |
| Recombinant DNA reagent | pE-SUMO-Odn* (plasmid) | This paper | pKS2189 | Vector to express 6xHisSumo-fusion protein, fused to N-terminus of Odn* (Gp88$^{E180A}$) |
| Strain, strain background (Vibrio cholerae) | PLE V. cholerae (E7946) | *Levine et al., 1982* | KDS6 | |
| Strain, strain background (Vibrio cholerae) | PLE1 V. cholerae (PLE1 E7946) | *O'Hara et al., 2017* | KDS36 | |
| Strain, strain background (Vibrio cholerae) | PLE2 V. cholerae (PLE2 E7946) | *O'Hara et al., 2017* | KDS37 | |
| Strain, strain background (Vibrio cholerae) | PLE3 V. cholerae (PLE3 E7946) | *O'Hara et al., 2017* | KDS38 | |
| Strain, strain background (Vibrio cholerae) | PLE4 V. cholerae (PLE4 E7946) | *O'Hara et al., 2017* | KDS39 | |
| Strain, strain background (Vibrio cholerae) | PLE5 V. cholerae (PLE5 E7946) | *O'Hara et al., 2017* | KDS40 | |

*Continued on next page*

*Continued*

| Reagent type (species) or resource | Designation | Source or reference | Identifiers | Additional information |
|---|---|---|---|---|
| Strain, strain background (*Vibrio cholerae*) | PLE1 Δori *V. cholerae* (PLE1 E7946) | This paper | KDS297 | Used for all spot assays |
| Strain, strain background (*Vibrio cholerae*) | PLE2 Δori *V. cholerae* (PLE2 E7946) | This paper | KDS298 | *Figure 3—figure supplement 2* |
| Strain, strain background (*Vibrio cholerae*) | PLE2 ΔrepA *V. cholerae* (PLE2 E7946) | This paper | KDS299 | *Figure 3—figure supplement 2* |
| Strain, strain background (*Vibrio cholerae*) | PLE1 Δ*repA* Δori::ori$^{PLE2}$; P$_{tac}$-repA$^{PLE1}$ *V. cholerae* E7946 | This paper | KDS300 | *Figure 3B* |
| Strain, strain background (*Vibrio cholerae*) | PLE1 Δ*repA* Δori::ori$^{PLE2}$; P$_{tac}$-repA$^{PLE2}$ *V. cholerae* E7946 | This paper | KDS301 | *Figure 3B* |
| Strain, strain background (*Vibrio cholerae*) | PLE2 Δ*repA* Δori::ori$^{PLE1}$; P$_{tac}$-repA$^{PLE1}$ *V. cholerae* E7946 | This paper | KDS302 | *Figure 3B* |
| Strain, strain background (*Vibrio cholerae*) | PLE2 Δ*repA* Δori::ori$^{PLE1}$; P$_{tac}$-repA$^{PLE2}$ *V. cholerae* E7946 | This paper | KDS303 | *Figure 3B* |
| Strain, strain background (*Vibrio cholerae*) | PLE4 Δori *V. cholerae* (PLE4 E7946) | This paper | KDS304 | Used for all spot assays |
| Strain, strain background (*Vibrio cholerae*) | PLE5 Δori *V. cholerae* (PLE5 E7946) | This paper | KDS305 | Used for all spot assays |
| Strain, strain background (*Vibrio cholerae*) | PLE1 Δiterons *V. cholerae* (PLE1 E7946) | *Barth et al., 2020b* | KDS263 | Used for all spot assays |
| Strain, strain background (*Vibrio cholerae*) | PLE2 Δori::ori $^{PLE1}$ *V. cholerae* (PLE2 E7946) | This paper | KDS306 | Used for all spot assays |
| Strain, strain background (*Vibrio cholerae*) | PLE1Δori::ori$^{Mut}$ Δ*lacZ*::KanR *V. cholerae* E7946 (referred to as PLE1$^{Mut}$) | This paper | KDS319 | Ori engineered to match what is observed in PLE1(+) strains from Pakistan: biosample accession numbers SAMN08979118, SAMN08979175, SAMN08979185, SAMN08979188, and SAMN08979253 |
| Strain, strain background (*Escherichia coli*) | pE-SUMO-Odn *E. coli* BL21 | This paper | KDS307 | Expression strain for Gp88/Odn |
| Strain, strain background (*Escherichia coli*) | pE-SUMO-Odn* *E. coli* BL21 | This paper | KDS308 | Expression strain for Gp88*/Odn* |
| Strain, strain background (Bacteriophage ICP1) | 2006 WT (ICP1_2006_Dha_E) | *O'Hara et al., 2017* | MH310934 | |
| Strain, strain background (Bacteriophage ICP1) | 2006 ΔCR; ΔCas2_3 (ICP1_2006_Dha_E) | *McKitterick and Seed, 2018* | | |

*Continued on next page*

*Continued*

| Reagent type (species) or resource | Designation | Source or reference | Identifiers | Additional information |
|---|---|---|---|---|
| Strain, strain background (Bacteriophage ICP1) | 2001 WT (ICP1_2001_Dha_0) | *Seed et al., 2011* | HQ641347 | |
| Strain, strain background (Bacteriophage ICP1) | 2001 Δ*odn* (ICP1_2001_Dha_0) | This paper | KSφ93 | *odn* is *gp88* |
| Strain, strain background (Bacteriophage ICP1) | 2001 *odn** (ICP1_2001_Dha_0) | This paper | KSφ134 | *odn** is *gp88*$^{E180A}$ |
| Sequence-based reagent | 5'-AGGGTTTGAGTGCGATTACG-3' | *O'Hara et al., 2017* | zac14 | qPCR primer targeting a conserved portion of the PLE noncoding region |
| Sequence-based reagent | 5'-TGAGGTTTTACCACCTTTTGC-3' | *O'Hara et al., 2017* | zac15 | qPCR primer targeting a conserved portion of the PLE noncoding region |
| Sequence-based reagent | 5'-GTCATTTAACGCATCTTATCACC-3' | This paper | KS459 | F-primer used to amplify noncoding region probes for PLE1 and PLE5 |
| Sequence-based reagent | 5'-GGCTTAGCAACTGTCTACGG-3' | This paper | zac267 | F-primer used to amplify noncoding region probes for PLE2, PLE3, and PLE4 |
| Sequence-based reagent | 5'-GTTACGTCTGATTGCTGACG-3' | This paper | KS321 | R-primer used to amplify noncoding region probes for PLE1 |
| Sequence-based reagent | 5'-CCGCTTATATCAATTTCACTAATATCT-3' | This paper | zac269 | R-primer used to amplify noncoding region probes for PLE2 and PLE3 |
| Sequence-based reagent | 5'-GGACGGCTAAACCATTCTCG-3' | This paper | KS323 | R-primer used to amplify noncoding region probes for PLE4 and PLE5 |
| Sequence-based reagent | 5'-CATAAGGTTGGCTCCTCAATG-3' | This paper | KS458 | R-primer used to amplify noncoding region probe for PLE4 in *Figure 4—figure supplement 5* |

## Strains and culture conditions

*V. cholerae* strains used in this study are derived from E7946. Bacteria were routinely grown on LB agar plates and in LB broth with aeration at 37°C. Antibiotics were supplemented as appropriate at the following concentrations: 75 µg/ml kanamycin, 100 µg/ml spectinomycin, 1.25 or 2.5 µg/ml chloramphenicol (*V. cholerae* for broth or plate conditions, respectively), 25 µg/ml chloramphenicol (*Escherichia coli*), and 100 µg/ml streptomycin. A detailed list of all strains used throughout this study can be found in the Key resources table.

Phage titers were determined using a soft agar overlay method wherein ICP1 was allowed to adsorb to *V. cholerae* for 10 minutes at room temperature before the mixture was added to molten

LB soft agar (0.5%) and poured onto 100 mm × 15 mm LB agar plates. Plaques were counted after overnight incubation at 37°C. Prior to phage infection for purposes of quantification or qPCR or spot assay analysis, *V. cholerae* was grown on plates overnight and then inoculated into 2 ml LB liquid cultures. Liquid cultures were grown to an OD > 1, then back diluted in fresh media to $OD_{600}$ = 0.05, and then grown to $OD_{600}$ = 0.3, at which point they were infected.

## Generation of mutant strains and constructs

*V. cholerae* mutants were generated through natural transformation as described previously (*Dalia et al., 2014*). For gene knockouts, splicing by overlap extension (SOE) PCR was used to generate deletion constructs with a spectinomycin resistance cassette flanked by frt recombination sites. Following selection of spectinomycin-resistant mutants, a plasmid bearing an isopropyl β-d-1-thiogalactopyranoside (IPTG)-inducible Flp recombinase was mated into transformants and Flp expression was induced to generate in-frame deletions. The plasmid was cured by growing mutants under inducing conditions with 300 µg/ml streptomycin. For unmarked replication origin-swapped constructs, mutants were generated through natural transformation by cotransformation (*Dalia et al., 2014*). For plasmid expression constructs, a derivative of the pMMB67EH vector with a theophylline-inducible riboswitch was used as previously described (*McKitterick and Seed, 2018*). All constructs were confirmed with DNA sequencing over the region of interest, and primer sequences and construct designs are available on DRYAD at https://datadryad.org/stash/share/HSB-bM3fCu3gSdF_yMQpCqyYuT4wW6_2IsZAkY0P5Ho.

## Phage infection spot assays

*V. cholerae* was added to molten 0.5% LB top agar and poured over LB plates. Following solidification of the top agar, 3 µl of serially 10-fold diluted phage were spotted onto the plate. Once phage spots dried, plates were incubated for at 37°C for 2 hr and then overnight at room temperature before visualization.

## Real-time quantitative PCR

qPCR experiments were performed as previously described (*Barth et al., 2020b*; *O'Hara et al., 2017*). Briefly, liquid cultures were infected with ICP1 at a multiplicity of infection (MOI) of 2.5 at $OD_{600}$ = 0.3. Samples were taken at 0 and 20 minutes post infection and boiled before serving as templates for IQ SYBR (Bio-Rad) qPCR reactions. For assays involving induction of *repA*, 2 ml cultures were grown with 1.25 µg/ml chloramphenicol for plasmid maintenance and induced for 20 minutes prior to infection using a final concentration of 1.5 mM theophylline and 1 mM IPTG starting at $OD_{600}$ = 0.17. All conditions were tested in biological triplicate, and each reported data point is the mean of two technical replicates. A single primer set (Key resources table) that amplifies a conserved region in all PLEs was used to detect PLE replication by qPCR.

## Protein purification

*E. coli* BL21 cells containing a $His_6$-SUMO fusion to WT or E185A Gp88 were grown to $OD_{600}$ = 0.5 at 37°C and induced with IPTG to a final concentration of 0.5 mM. The culture was grown for 2 hr and harvested by centrifugation at 4000×g for 20 minutes. The pellet was resuspended in lysis buffer (50 mM Tris–HCl pH 8, 200 mM NaCl, 1 mM BME, 0.5% Triton-X 50 mM imidazole, 1 Pierce Protease Inhibitor Mini Tablet [Thermo Scientific]) and sonicated. Cell debris was removed by centrifugation (29,097×g for 40 minutes). The lysate was applied to a HisTrap HP column (Cytiva). The column was washed with wash buffer (50 mM Tris–HCl pH 8, 200 mM NaCl, 1 mM BME, 50 mM imidazole), and a high salt wash (50 mM Tris–HCl pH 8, 2 M NaCl, 1 mM BME, 50 mM imidazole) was used to remove residual DNA. The protein was eluted using an elution buffer (50 mM Tris–HCl pH 8, 200 mM NaCl, 1 mM BME, 300 mM imidazole), and then the eluate was applied to a HiTrap Heparin HP column (Cytivia) for further purification. Following elution from the HiTrap Heparin column, the protein was dialyzed using a 10k Slide-A-Lyzer Dialysis cassette (Thermo Fisher) in 50 mM Tris–HCl pH 7.5, 150 mM NaCl, 1 mM dithiothreitol (DTT). Concomitant with dialysis, the $His_6$-SUMO tag was cleaved using SUMO protease. The SUMO tag was removed using Dynabeads (Invitrogen).

## Sequence analysis

All genomes were visualized and compared in CLC Main Workbench 7. Multiple sequence alignments were performed using the Multiple Sequence Alignment (MUSCLE) tool with default settings (*Edgar, 2004*). The phylogenetic tree was constructed using the IQ-TREE web interface with default settings (*Trifinopoulos et al., 2016*). Conservation of PLE sequence was compared and visualized using Mauve (*Darling et al., 2004*).

## Nuclease assays

Nuclease assays were performed with 100 ng of DNA probes and up to 500 nM of purified Gp88 in 20 µl reactions with 50 mM Tris, 10 mM $MgCl_2$, 50 mM NaCl, 1 mM DTT reaction buffer. Reactions proceeded at 30°C for 30 minutes, and were visualized on 0.8% agarose gels ran at 80 V for 30 minutes, and stained with GelRed (Biotium). For smaller probes (*Figure 6*), 25 ng of probe was included in reactions, and the product was visualized on 2% agarose gels ran at 120 V for 20 minutes and stained with GelGreen (Biotium). Primers used for probe amplification can be found in the Key resources table.

## Acknowledgements

This work was supported by the National Institute of Allergy and Infectious Diseases (grant numbers R01AI127652 and R01AI153303 to KDS); KDS is a Chan Zuckerberg Biohub Investigator and holds an Investigators in the Pathogenesis of Infectious Disease Award from the Burroughs Wellcome Fund. We would like to thank Yue Clare Lou for construction of strains, as well as Stephanie Hays, Kristen LeGault, and Zoe Netter for critical reading of the manuscript and providing useful advice.

## Additional information

### Competing interests

Kimberley D Seed: is a scientific advisor for Nextbiotics, Inc. The other authors declare that no competing interests exist.

### Funding

| Funder | Grant reference number | Author |
| --- | --- | --- |
| National Institute of Allergy and Infectious Diseases | R01AI127652 | Kimberley D Seed |
| National Institute of Allergy and Infectious Diseases | R01AI153303 | Kimberley D Seed |
| Burroughs Wellcome Fund | 1019213 | Kimberley D Seed |
| Chan Zuckerberg Initiative | | Kimberley D Seed |

The funders had no role in study design, data collection and interpretation, or the decision to submit the work for publication.

### Author contributions

Zachary K Barth, Conceptualization, Formal analysis, Investigation, Writing - original draft; Maria HT Nguyen, Investigation, Writing - review and editing; Kimberley D Seed, Conceptualization, Supervision, Funding acquisition, Investigation, Writing - review and editing

### Author ORCIDs

Zachary K Barth ⬚ https://orcid.org/0000-0002-1321-306X
Maria HT Nguyen ⬚ https://orcid.org/0000-0003-2441-4948
Kimberley D Seed ⬚ https://orcid.org/0000-0002-0139-1600

Decision letter and Author response
Decision letter https://doi.org/10.7554/eLife.68339.sa1
Author response https://doi.org/10.7554/eLife.68339.sa2

## Additional files

### Supplementary files
• Transparent reporting form

### Data availability

All data generated or analyzed during this study are included in the manuscript and supporting files. Design for genetic constructs including primers are available via Dyrad (https://doi.org/10.6078/D1T704).

The following dataset was generated:

| Author(s) | Year | Dataset title | Dataset URL | Database and Identifier |
|---|---|---|---|---|
| Barth ZK, Nguyen MHT, Seed KD | 2021 | Strains and constructs for: A chimeric nuclease substitutes a phage CRISPR-Cas system to provide sequence specific immunity against subviral parasites | https://doi.org/10.6078/D1T704 | Dryad Digital Repository, 10.6078/D1T704 |

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
