## [Decision Letter]

**Acceptance summary:**

Mobile genetic parasites in *Vibrio cholerae* genomes often protect the host from phage infection. Previous work has shown that some phages that infect *Vibrio cholerae* have evolved CRISPR-systems that selectively target these defensive mobile genetic elements and thereby restore infection by the phage. Here the authors show that phage genomes missing the CRISPR-system, often rely on a chimeric nuclease that functionally replace the CRISPR-system, revealing an ongoing evolutionary battle that includes one host and two competing parasites.

**Decision letter after peer review:**

Thank you for submitting your article "A chimeric nuclease substitutes CRISPR-Cas: A phage weaponizes laterally acquired specificity to destroy subviral parasites" for consideration by *eLife*. Your article has been reviewed by 3 peer reviewers, and the evaluation has been overseen by Blake Wiedenheft as the Reviewing Editor and Gisela Storz as the Senior Editor. The following individuals involved in review of your submission have agreed to reveal their identity: Luciano Marraffini (Reviewer #2); Edze Westra (Reviewer #3).

Essential revisions:

1) Streamline the introduction to include only information critical to understanding how your work advances the field.

2) An SDS-PAGE of the purified WT and mutant proteins is required to demonstrate purity and help support the conclusion that the Odn nuclease alone is responsible for cleavage of the PLE.

3) Clarify inconsistencies and variabilities observed in the nuclease activity assays. For example, Figure 4d shows PLEs 1, 4, and 5 are cleaved to completion by WT Odn, while in Figure 6c and its supplements, the WT shows very little cleavage activity against PLE1. Different lengths of the substrates will shift the molar ratio of enzyme to substrate, and this impact the nuclease activity. Cleavage efficiencies must be quantified and normalized.

4) Perform plaque assays for the ori mutant that renders this PLE resistant to Odn Figure 6c. This will be similar to the presentation of data in Figure 5a.

5) Please specify that the evidence for Odn activity driving evolutionary change in the PLE is anecdotal.

6) Sequence Odn cleavage products and map the precise location of these cuts.

*Reviewer #1:*

Barth et al. discover and characterize a new mechanism by which bacteriophage ICP1 defends itself against its own molecular parasites, the PLEs. PLEs are short stretches of DNA found in ~15% of sequenced epidemic strains of V. cholerae which hijack the structural components of bacteriophage ICP1 to facilitate their own spread-through this process, PLEs promote survival of the bacteria by preventing the spread of an ICP1 infection. Therefore, the interplay between ICP1 and its PLEs have significant impacts on Vibrio pathogenesis, highlighting the importance of this work. Previous research from this group has shown that some ICP1 variants harbor a CRISPR-Cas immune system, which these bacteriophages use to target and degrade PLEs. However, not all ICP1 variants possess CRISPR-Cas systems, and yet retain resistance to some PLE variants, suggesting the presence of other anti-PLE immune systems encoded in the bacteriophage genome. In this study, the investigators seek to understand how an ICP1 variant devoid of a CRISPR-Cas system can still defend against a subset of PLEs.

Strengths

A rigorous set of genetics experiments were applied to reveal conserved functional aspects of PLE replication (Figure 3 and its supplements) as well as the genetic element in ICP1 required for anti-PLE activity, called gp88 (Figures 1, 2, and associated supplements). Based on sequence alignments and preliminary biochemical characterization, the gp88 gene product was shown to facilitate nuclease activity against PLE variants that are sensitive to ICP1 (Figure 4d). Genetics was also successfully applied to narrow down the region within the PLE which is targeted by the gp88 (Odn) nuclease (Figures 4a-c and 5). Finally, a natural variant of PLE1 that seems resistant to Odn activity was discovered within a V. cholerae isolate from Pakistan (Figure 6), providing a beautiful example of how the ongoing conflict between host and parasite drives genetic diversification of both parties. Overall, the major conclusions in the paper are well-supported by the data, the manuscript is clearly written, and the study gives important insights into this system.

Weaknesses

The biochemical characterization of Odn is preliminary and needs more rigorous experimentation.

Comments for the authors:

1. An image of the purified proteins, both WT and mutant variants, (typically in the form of SDS-PAGE gels), should be displayed to demonstrate their purity and help support the conclusion that the Odn nuclease alone is responsible for cleavage of the PLE. The methods describe three separate chromatography steps, so it is assumed that the proteins are relatively pure, but a gel should be shown.

2. The mutant version of Odn should be subject to the same level of scrutiny as the WT to solidify the conclusion that Odn is alone responsible for PLE cleavage, at least in the beginning. For example similar assays could be performed with the mutant as with WT in Figures 4c and Figure 5a.

3. There seems to be some inconsistencies/variabilities in the biochemical assays. For example, Figure 4d shows PLEs 1, 4, and 5 are cleaved to completion by WT Odn, while in Figure 6c and its supplements, the WT shows very little cleavage activity against PLE1. Some quantification will help the reader to understand the extent of the variation in the nuclease activity. Along the same lines, different lengths of substrates were used in these assays, which will shift the molar ratio of enzyme:substrate and impact the nuclease activity. Some sort of normalization across the assays will allow the reader to compare the levels of nuclease activity from gel to gel and assess how each new variable affects the activity.

4. To come full-circle, sequencing of the cut products would go a long way to support the genetics evidence for Odn targeting within the PLE origin and map the precise location of the cut(s).

*Reviewer #2:*

Barth et al. report the discovery of a novel endonuclease effector employed by certain lytic vibriophages (ICP1) to overcome viral satellites called PLEs present in ~15% of sequenced epidemic *Vibrio cholerae*. Past work from the Seed lab (Seed et al., 2013) had demonstrated that some ICP1 isolates (especially those from 2011-2017) encoded their own type I-F CRISPR-Cas system to target PLEs. However, they also previously observed (O'Hara et al., 2017) that older ICP1 isolates that lacked CRISPR-Cas (2001-2011) could still plaque on some of the strains containing PLEs. This work sought to elucidate the mechanisms by which a CRISPR-less ICP1 isolate from 2001 could successfully overcome certain PLE variants, but not others.

Through a set of simple yet elegant experiments that directly tested their hypotheses, the authors convincingly show that ICP12001 encodes an endonuclease (initially gp88, which they later term Odn, for origin-directed nuclease) that allows it to successfully infect V. cholerae containing certain PLEs. Interestingly, Odn contains two domains-the C-terminus resembling a homing endonuclease (HEG), and the N-terminus resembling the DNA-binding domain of the replication initiator RepA of PLEs 1, 4, and 5. The authors speculate that Odn might have arisen from domain shuffling between ICP1-encoded HEGs and PLE-encoded RepA, although how exactly this occurred is unknown.

The major strengths of this work are the novelty of the findings and its logical clarity. With a series of incisive experiments, the authors deduced how Odn works. By constructing ICP1 mutants that lacked Odn activity, the authors showed that Odn is necessary for plaque formation on V. cholerae containing PLEs 1, 4, or 5. Further, they show using qPCR that plaquing efficiency on PLE1 is correlated with Odn-mediated repression of PLE1 replication. In the future, it will be interesting to know if Odn alone is sufficient to overcome PLE, and whether the adjacent genes broN and gp90 play any regulatory or accessory role in Odn function.

Next, the authors compared the genomes of all five PLE variants and found that the ori and RepA sequences cluster into two distinct groups-PLEs 1/4/5 and PLEs 2/3. This potentially explained why ICP12001 can plaque on strains containing PLEs 1, 4, and 5, but not PLEs 2 or 3, leading the authors to hypothesize that Odn recognizes and binds to the origin of replication of PLEs 1/4/5, but not PLEs 2/3. The Seed lab previously showed that non-replicating PLE mutants lacking ori could still restrict ICP1 (Barth et al., 2020). Exploiting this experimental tool, they showed here that ICP12001 could no longer plaque on strains containing PLE mutants lacking ori. Finally, by re-constituting the ori of PLE1 into PLE2, the strain is now sensitized to Odn activity upon ICP12001 infection. This decisive result strongly suggests that recognition of ori sequences is required for Odn-mediated plaquing.

Finally, using an in vitro cleavage assay, the authors show that purified Odn cleaves PCR products amplified from the non-coding region containing the ori from PLEs 1/4/5, but not PLEs 2/3, consistent with the in silico prediction that the C-terminus of Odn resembles putative HEGs in ICP1. The authors further narrow down the region of specificity of Odn action to a set of iterons (a series of three ~30bp semi-palindromic repeats in the origin of replication) and demonstrate using a PLE1 mutant lacking these sequences that iterons are necessary for Odn-mediated cleavage and plaquing by ICP12001. Surprisingly, there is only a 2-bp difference between PLEs 4 and 5, yet the cleavage pattern looks different between the two, raising the question of whether a more complicated mechanism of recognition and cleavage is in place.

Finally, the authors address a crucial aspect of any host-parasite arms race-how can PLE escape from Odn-mediated restriction? They reason that simple deletions of the replication origin would not likely be selected for as a long-term solution, given its essentiality for PLE reproduction. Because of this, the authors speculate that Odn selected for domain shuffling events in PLE RepA, resulting in the two distinct clusters of PLEs 1/4/5 and PLEs 2/3. Although each PLE variant is remarkably conserved, the authors found an instance where PLE1 contained variation in its iteron sequences. They tested this variant and found that Odn was unable to cleave, suggesting that this variant PLE1 was likely resistant to infection by ICP12001, although they did not perform a plaque assay to confirm this.

In conclusion, the authors complete a thorough characterization of Odn as an anti-PLE effector in ICP1. This work presents a key advance in our understanding of ICP1-PLE dynamics and brings broader insight to the roles of lateral gene transfer in the acquisition of novel weaponry in the host-parasite arms-race.

Comments for the authors:

Experimental:

Figure 1: can Odn be re-constituted into a CRISPR-less ICP1 and restore plaquing against PLE1 to demonstrate sufficiency?

Figure 6: In addition to the cleavage assay, to complement the in vitro data with in vivo results, the authors should perform plaque assays on PLE1mut for ICP12001 (and the Odn deletion and catalytically inactive mutants). It would be expected for ICP12001 to no longer be able to plaque given that Odn cannot cleave.

Optional: it would be interesting to see if PLE escapers could arise naturally during an infection experiment and to determine what these mutations were. For example, you can infect a strain containing PLE1 with ICP12001 at very high MOI to lyse a whole plate. Are there any bacterial-insensitive mutants (BIMs) that form colonies? If so, are there any mutations in ori (and/or compensatory mutations in RepA), or perhaps even more strikingly, any restructuring of the iteron? This experimental evidence, together with the naturally occurring PLE1mut variant observed in Figure 6, would bolster the claim that Odn may be a driving force for the diversification of the PLE replication origin.

Text edits:

Title: the title, although descriptive, is a bit long and convoluted without having read the paper first. I would consider simplifying it to a single clause and remove jargon like "laterally acquired specificity"

Introduction: the description of MGE and HGT in multicellular organisms in the first paragraph is not necessary to understand this work and possibly too much information for the reader.

Figure 2: although this is beyond the focus of this work and does not need to be addressed in the text, it does not escape my attention that PLE3 is able to evade both CRISPR (no targeting spacer) and Odn (ori not recognized). Is there a contemporaneous ICP1 isolate that is able to plaque on PLE3, and if so, is there a different effector in the same genomic region between gp87 and gp91?

Figure 3: is there a working model for how PLEs recombine (if they even do) to form new variants? Are their isolates of V. cholerae that contain multiple PLEs?

Lines 342-343: "Remarkably, Gp88's own N-terminal domain is 61% similar to 93% of PLE1's RepA_N domain".

This sentence is a bit misleading, given that it's only 42% identical (which is still quite similar). The inclusion of "similar residues" in the comparison indeed yields 61%, but this factor in the calculation should be explicitly stated in the text.

Figure 4D: There appears to be two cuts for PLEs 1 and 5, but not 4? This is interesting, especially since there's only a 2-bp difference in the iteron sequences between PLEs 4 and 5 (Figure 6)-is there any explanation for the differences between the cleavage pattern of the three PLEs at this time, and whether these provide any clues as to Odn's mechanism of action?

Discussion:

Lines 551-552: what are some examples of auxillary genes that are present between CRISPR-less ICP1 vs. CRISPR(+) ICP1? Are CRISPR-less ICP1 isolates on average just slightly smaller?

*Reviewer #3:*

Phage ICP1 infection can be blocked by PLE elements in Vibrio genomes, which use phage capsids for their own dissemination, but ICP1 has in turn evolved mechanisms to overcome PLE-mediated defence/parasitism. Seed and colleagues have discovered and extensively characterised how a CRISPR-Cas system encoded by the phage can cleave the PLE element, leading to productive infections.

Here, the authors aimed to explain why an ICP1 genotype that naturally lacks CRISPR-Cas, and that used to be very common before CRISPR-carrying variants appeared, can infect Vibrio cells that certain PLE variants. Specifically, their previous work found that this ICP1 variant can infect cells that carry the two oldest PLE variants, PLE5 and PLE4, as well as the most recent variant PLE1 (O'Hara et al., 2017).

They found that this older ICP1 genotype carries an alternative nuclease (termed origin-directed nuclease, Odn) that cleaves the origin of replication of some PLE elements. This nuclease gene is located in the same locus as the CRISPR-Cas genes in more recent ICP1 genotypes. This newly discovered nuclease carries an N-terminal DNA binding domain that has high sequence similarity to RepA, which binds the origin of replication, and a C-terminal nuclease domain that is similar to a ICP1-encoded family of putative homing endonucleases (HEGs), and that is responsible for cleavage of the origin of replication. Different PLE variants (PLE1-5) vary with respect to their RepA and origin of replication sequences – specifically, the authors show that PLE1, 4 and 5 share RepN and Ori sequences, explaining why the ICP1 variant that carries Odn can infect cells that carry these PLE elements, but not cells that carry PLE2 or PLE3.

The conclusions of this paper are well supported by comparative genomics, elegant reverse genetics analyses and biochemical assays. The authors generate all the relevant KO variants of ICP1 and PLE to demonstrate the genetic interaction between Odn of ICP1 and the Ori of PLE, including ori-swapped mutants of PLE, where PLE2 carries the ori of PLE1, leading to a lack of protection of the host from phage infection. Finally, the authors purified Odn and demonstrate that it can cleave the Ori sequences in vitro, but not random sequence, and a catalytic mutant is unable to cleave either (and the same mutation leads to PLE sensitivity in vivo). This binding and cleavage is specifically directed towards a set of semi-palindromic sequences, known as iterons.

All in all, this is a very thorough and complete study that provides novel and deep insight into the ongoing coevolutionary interaction between PLE and ICP1.

Comments for the authors:

This is a very nice study – I enjoyed reading the paper. I only have fairly minor comments/suggestions:

1. This is a matter of preference in style, but in my opinion the introduction and discussion could be much tighter. The first 3 paragraphs of the introduction provide a lot of background that is not essential for this study, perhaps more appropriate for a broad review, and could be trimmed substantially. The discussion is

2. I'm left wondering whether BroN and gp90 are essential for the activity of Odn. Have you made knockouts of these genes ? Or alternatively, have you expressed Odn from a plasmid in cells with PLE and looked for evidence of cleavage (e.g. cytotoxic effects that are not observed for the catalytic mutant) ? You mention they likely have regulatory roles – what is the evidence in support of this idea ?

3. Figure 3A shown the ori and RepA-N sequence variation between PLE1,4,5 on the one hand, and PLE2,3 on the other. Supplement 1 only shows the alignment of the conserved c-terminus. Can you please add the N-terminal sequence to enable more detailed examination of the level of sequence variation / conservation between the different groups?

4. the similarity of the nuclease domain to the putative homing endonucleases in ICP1 is intriguing, and provides a plausible explanation for the origin of the nuclease domain. How common are these homing endonucleases? Can the authors make a tree of these nuclease domains to see if this Odn nuclease domain fall inside the tree? Or do you not have enough sequences for this / is sequence divergence too high?

5. Figure 4 D – nice to see the cleavage activity of gp88/Odn. It should not be too hard to identify the cleavage site using Sanger sequencing (it looks like sharp bands, so i assume a single cut). Have you tried this?

6. Line 456-459 – I don't think the data are sufficient to draw this conclusion; even if the mutation provides resistance to Odn, it is impossible to work out from these data whether or not the Ori mutations are adaptive, and if so, whether or not this is because of ICP1 infections. This statement therefore needs to be further toned down in my opinion.

7. Line 544 – the Gomez et al. Science paper from the same group would also be appropriate to cite.

---

## [Author Response]

Essential revisions:1) Streamline the introduction to include only information critical to understanding how your work advances the field.

We have streamlined the introduction removing several examples.

2) An SDS-PAGE of the purified WT and mutant proteins is required to demonstrate purity and help support the conclusion that the Odn nuclease alone is responsible for cleavage of the PLE.

We have added images of the SDS-PAGE gels of protein preparations used in in vitro assays as Figure 4—figure supplement 2.

3) Clarify inconsistencies and variabilities observed in the nuclease activity assays. For example, Figure 4d shows PLEs 1, 4, and 5 are cleaved to completion by WT Odn, while in Figure 6c and its supplements, the WT shows very little cleavage activity against PLE1. Different lengths of the substrates will shift the molar ratio of enzyme to substrate, and this impact the nuclease activity. Cleavage efficiencies must be quantified and normalized.

For Figure 6C we originally used a shorter substrate for both the WT PLE1 origin andthe mutant PLE1 origin because we had difficulty cloning the mutant origin sequence into the larger PLE1 context (since the mutant was identified in sequenced strains from other groups and not in our collection). We suspect you are correct that the loss of cutting was due to a decreased molar ratio of enzyme to substrate. We have now successfully generated the longer mutant PLE1 origin probe and have repeated the experiment using probes of the same length as in Figure 4D. These new data are shown in revised Figure 6C (replicate cleavage assays are shown in Figure 6 Figure supplement 1). With the longer probe we do see some cleavage of the PLE1^Mut^ at high concentrations of Odn, however, together with additional in vivo evidence (discussed in response to point 4), our data indicate that the mutant ori is indeed resistant to Odn-mediated cleavage.

We also want to point out that we do not consistently see complete loss of the uncleaved substrate of even the same probe, for example, compare the three replicates of Figure 4C (Figure 4 supplement 3) – we only saw complete loss of the uncleaved substrate for the PLE1 probe in one replicate. We have therefore replaced the gel in the main figure with one of the (former) supplementary gels to reflect that complete cleavage is not the outcome the majority of the time. We are unfortunately not able to explain this minor inconsistency, and given that we make no claims regarding cleavage efficiency in vitro (and do not feel comfortable doing so), we did not quantify cleavage efficiency. We hope that the combination of in vivo and in vitro evidence, which together tell the same story regarding cleavage specificity, is satisfactory.

4) Perform plaque assays for the ori mutant that renders this PLE resistant to Odn Figure 6c. This will be similar to the presentation of data in Figure 5a.

We have now completed these experiments, and the results are presented in Figure 6E. These data show that PLE1^Mut^ is resistant to Odn(+) ICP1 as anticipated. To take this one step further, we also performed qPCR of the PLE1^Mut^ strain during ICP1 infection and show that PLE1^Mut^ replicates during infection and the copy increase is not impacted by Odn (Figure 6D). Collectively, we feel these new data provide more support for our conclusion that this mutation is adaptive in the face of Odn.

5) Please specify that the evidence for Odn activity driving evolutionary change in the PLE is anecdotal.

We have addressed this point: we have removed our original assertion that “These data

suggest that Odn imposes selective pressure on the origin of replication…” and substituted more cautious language, particularly with regard to the data in revised Figure 6: “These data demonstrate that PLE can escape Odn activity through subtle restructuring of the iterons, in addition to more extensive replication module exchange (Figure 3A), and suggest that ICP1 defenses like Odn may select for diversification of the PLE replication machinery.”

As well as in the discussion: “PLEs are able to escape Odn antagonism through mutation of their origin of replication and there is compelling evidence suggesting that PLEs have exchanged replicon modules for an alternative replication origin and initiation factor origin binding domain on at least two separate occasions.”

We feel that we were also appropriately cautious in the language we used in parts of the

original submission (these examples remain unchanged in the revised manuscript):

“The endonuclease appears to exert considerable selective pressure on PLEs

and may drive PLE replication module swapping”;

“It is possible that Odn selected for alternative replicon modules, leading to the

decline of PLE4 and succession of PLE2 that occurred in the early 2000s (O’Hara et al., 2017). […] This seems like a likely possibility for PLEs and ICP1 whose specific adaptations.”

6) Sequence Odn cleavage products and map the precise location of these cuts.

This was something we tried (several times) prior to the initial submission, and in response to reviewer comments made several more attempts. Using Sanger sequencing of Odn cleavage products we were unable to make a clear determination of where cutting was taking place. In most cases we saw uncleaved substrate in our sequencing results (see response to point 3), even when no uncleaved product was visible on the gel and after gel extracting smaller cleaved products. Using higher protein concentrations and longer incubation times did not resolve this issue.

In some of our attempts to address this point, chromatograms obtained by Sanger sequencing did suggest that Odn cleavage of PLE1 is occurring proximal to the poly-T centers of the first and second iterons, but these sequencing results were somewhat inconsistent, and we do not have a robust method of calling sub-sequence ends from the amplitudes of base signals. We therefore do not feel confident enough to include these results in the manuscript.

We also attempted subcloning of cleavage products via TA cloning and blunt end cloning, but were unable to recover cleaved products through these methods.

We have, however, taken additional steps requested by reviewer 1 – including cleavage assays with Odn* and other substrates (see below). Ultimately, although we do not know the precise location where cleavage is taking place, we feel that our in vivo and in vitro results are sufficient to demonstrate Odn targeting through the iterons. We agree that our biochemical characterization of Odn is preliminary and indeed include that this is an area that needs further study: “however the molecular details of Odn binding and catalysis remain to be elucidated.”

Reviewer #1:[…] Comments for the authors:1. An image of the purified proteins, both WT and mutant variants, (typically in the form of SDS-PAGE gels), should be displayed to demonstrate their purity and help support the conclusion that the Odn nuclease alone is responsible for cleavage of the PLE. The methods describe three separate chromatography steps, so it is assumed that the proteins are relatively pure, but a gel should be shown.

We have added images of the purified protein as Figure 4 —figure supplement 2.

2. The mutant version of Odn should be subject to the same level of scrutiny as the WT to solidify the conclusion that Odn is alone responsible for PLE cleavage, at least in the beginning. For example similar assays could be performed with the mutant as with WT in Figures 4c and Figure 5a.

In figure 5A, we include Odn* at our maximum concentration of 500nM and detect no cleavage of the PLE1 probe. In response to this comment we have now tested Odn* against other probes (PLE4 and PLE 5) which were cut by Odn in Figure 4D. These new data are presented in Figure 4E (replicate assays are found in Figure 4—figure supplement 4) and show that 500nM Odn* does not cleave probes cut by Odn, consistent with the in vivo data and our conclusions.

3. There seems to be some inconsistencies/variabilities in the biochemical assays. For example, Figure 4d shows PLEs 1, 4, and 5 are cleaved to completion by WT Odn, while in Figure 6c and its supplements, the WT shows very little cleavage activity against PLE1. Some quantification will help the reader to understand the extent of the variation in the nuclease activity. Along the same lines, different lengths of substrates were used in these assays, which will shift the molar ratio of enzyme:substrate and impact the nuclease activity. Some sort of normalization across the assays will allow the reader to compare the levels of nuclease activity from gel to gel and assess how each new variable affects the activity.

Thank you for this suggestion. Please see our response to essential revisions point 3.

4. To come full-circle, sequencing of the cut products would go a long way to support the genetics evidence for Odn targeting within the PLE origin and map the precise location of the cut(s).

Please see our response to essential revisions point 6.

Reviewer #2:[…] Comments for the authors:Experimental:Figure 1: can Odn be re-constituted into a CRISPR-less ICP1 and restore plaquing against PLE1 to demonstrate sufficiency?

Thank you for the suggestion, reconstitution of Odn restoring ICP1 ∆CRISPR plaquing on PLE1 would strongly support our model and was in fact something we tried quite early on. Unfortunately we found that Odn in its native sequence context is too toxic to provide as an ICP1 editing template in *V. cholerae* (which requires cloning and subsequent maintenance in *E. coli* and *V. cholerae* to generate the genome editing strain) so we have not been able to reconstitute its presence within the ICP1 genome.

Figure 6: In addition to the cleavage assay, to complement the in vitro data with in vivo results, the authors should perform plaque assays on PLE1mut for ICP12001 (and the Odn deletion and catalytically inactive mutants). It would be expected for ICP12001 to no longer be able to plaque given that Odn cannot cleave.

Thank you for this suggestion. We have now performed these experiments and the results are presented in Figure 6E.

Optional: it would be interesting to see if PLE escapers could arise naturally during an infection experiment and to determine what these mutations were. For example, you can infect a strain containing PLE1 with ICP12001 at very high MOI to lyse a whole plate. Are there any bacterial-insensitive mutants (BIMs) that form colonies? If so, are there any mutations in ori (and/or compensatory mutations in RepA), or perhaps even more strikingly, any restructuring of the iteron? This experimental evidence, together with the naturally occurring PLE1mut variant observed in Figure 6, would bolster the claim that Odn may be a driving force for the diversification of the PLE replication origin.

This would indeed be very interesting. We did not attempt this; in our experience so far BIMs arising following infection of *V. cholerae* with ICP1 are overwhelmingly receptor loss mutants (Seed et al. PLOS Pathogens 2012). Further, while a selection method for PLE escape would be highly useful, the outcome of either ICP1 ‘winning’ or PLE ‘winning’ is cell death, which makes a selection of this sorts extremely challenging: PLE induction ultimately culminates in cell lysis, so even if PLE escape mutants were present, the bacteria harboring those mutant PLEs cannot be recovered. It is possible that PLE escapes (in the form of transductants) could be isolated, though we have previously shown that this does not happen with CRISPR(+) ICP1 (McKitterick et al. 2019), so we did not attempt it for Odn.

Text edits:Title: the title, although descriptive, is a bit long and convoluted without having read the paper first. I would consider simplifying it to a single clause and remove jargon like "laterally acquired specificity"

Thank you for the suggestion. We have changed the title to: “A chimeric nuclease substitutes a phage CRISPR-Cas system to provide sequence specific immunity against subviral parasites.”

We think this new title retains the information we wanted to convey while being more accessible.

Introduction: the description of MGE and HGT in multicellular organisms in the first paragraph is not necessary to understand this work and possibly too much information for the reader.

These examples, as well as few prokaryotic examples, have been removed to streamline the introduction.

Figure 2: although this is beyond the focus of this work and does not need to be addressed in the text, it does not escape my attention that PLE3 is able to evade both CRISPR (no targeting spacer) and Odn (ori not recognized). Is there a contemporaneous ICP1 isolate that is able to plaque on PLE3, and if so, is there a different effector in the same genomic region between gp87 and gp91?

Previously we’ve shown that spacers against PLE 3 are functional when acquired by this phage isolate (O’Hara et al.) The original isolate just happens to not have a spacer directed against PLE3. As for why this is the case, a possible explanation is that PLE3 appears to be much rarer than other PLEs, occurring in a relatively small number of sequenced *V. cholerae* strains.

We have not yet seen anything besides gp88 or CRISPR-Cas in the region between *gp87* and *gp91* in sequenced ICP1 isolates – this analysis includes an expanded set of ICP1 isolates collected between 1992-2019 and is part of a review article in press (Boyd CM, Angermeyer A, Hays SG, Barth ZK, Patel KM and Seed KD. 2021. Bacteriophage ICP1: A persistent predator of *Vibrio cholerae.* Annual Reviews of Virology). Of course we would be very intrigued to find isolates with different genes in that region of the ICP1 genome – we will see what we find in future sampling efforts.

Figure 3: is there a working model for how PLEs recombine (if they even do) to form new variants? Are their isolates of V. cholerae that contain multiple PLEs?

The mosaic sequences of PLEs suggest that PLEs have recombined at some point, but we do not have direct evidence for this recombination taking place. To date, no sequenced isolates have been found to harbor multiple PLEs. Potential explanations for this could be that carriage of multiple PLEs is unstable, or that PLEs recombine following transduction of a PLE into a strain that already has a PLE.

Lines 342-343: "Remarkably, Gp88's own N-terminal domain is 61% similar to 93% of PLE1's RepA_N domain".This sentence is a bit misleading, given that it's only 42% identical (which is still quite similar). The inclusion of "similar residues" in the comparison indeed yields 61%, but this factor in the calculation should be explicitly stated in the text.

We’ve amended this line to say "Remarkably, Gp88's own N-terminal domain is 42% identical and 61% sequence similar to 93% of PLE1's RepA_N domain". (P5 Line 233)

Figure 4D: There appears to be two cuts for PLEs 1 and 5, but not 4? This is interesting, especially since there's only a 2-bp difference in the iteron sequences between PLEs 4 and 5 (Figure 6)-is there any explanation for the differences between the cleavage pattern of the three PLEs at this time, and whether these provide any clues as to Odn's mechanism of action?

We apologize for any confusion regarding cleavage patterns. While it is true that there is only a 2-bp difference in the iteron sequence between PLEs 4 and 5, the probes used for our cleavage assays encompass a larger sequence harboring the origin of replication and contain numerous differences between the PLEs (the differences can be visualized in Figure 3A). Much of these differences consist of gain or loss of novel sequence. We designed our probes without considerable attention to where the iterons fell within the probes, and the PLE4 probe happened to have the iterons directly in its center. This is predicted to produce two cleavage products of the same size, which would appear as a single band on a gel (consistent with what is observed in Figure 4D).

We realize that this is likely to be a point of confusion for readers so we now address this in the text (P5 lines 265-270) and have added a supplementary figure (Figure 4—figure supplement 5), showing that when different primers are used to amplify the PLE4 probe (resulting in the iteron sequence offset from the center of the probe) the cleavage products clearly run as two bands, as would be predicted.

Discussion:Lines 551-552: what are some examples of auxillary genes that are present between CRISPR-less ICP1 vs. CRISPR(+) ICP1? Are CRISPR-less ICP1 isolates on average just slightly smaller?

We discuss auxiliary gene diversity and total genome length across ICP1 isolates in a forthcoming review article that’s been accepted for publication (Boyd et al. Annual Reviews of Virology, in press).

In summary, ICP1 isolates range in genome length from ~121kb to ~131kb, and the CRISPR-Cas system is ~7kb. Generally speaking yes, the isolates with the larger genomes tend to be CRISPR(+), but surprisingly, some of the largest ICP1 genomes lack CRISPR-Cas.

Reviewer #3:[…] Comments for the authors:This is a very nice study – I enjoyed reading the paper. I only have fairly minor comments/suggestions:1. This is a matter of preference in style, but in my opinion the introduction and discussion could be much tighter. The first 3 paragraphs of the introduction provide a lot of background that is not essential for this study, perhaps more appropriate for a broad review, and could be trimmed substantially.

We have streamlined the introduction.

2. I'm left wondering whether BroN and gp90 are essential for the activity of Odn. Have you made knockouts of these genes? Or alternatively, have you expressed Odn from a plasmid in cells with PLE and looked for evidence of cleavage (e.g. cytotoxic effects that are not observed for the catalytic mutant)? You mention they likely have regulatory roles – what is the evidence in support of this idea?

We have noted that WT Odn is cytotoxic in PLE1(+) hosts, but felt that the in vitro cleavage assays and infection host range phenotypes were more compelling than this toxicity phenotype and did not pursue it further.

While investigating the roles of the *broN*-related gene and *gp90* could yield interesting results, we do not think there’s enough space in this manuscript to also characterize their function.

In regard to these genes, we write “A gene encoding a Bro-N domain (pfam02498) and a KilAC domain (pfam03374) occurs adjacent to *gp88*. Their positions and putative annotations suggest that these divergently transcribed genes may have a regulatory function.” We believe our language is more cautious than saying that they are likely regulators, and merely puts forward a model consistent with their annotations, location, and orientation. It is common for regulators to be divergently transcribed from the genes they control. The BroN domain is thought to be a DNA binding domain, and the KilAC domain is associated with transcriptional antitermination activity.

3. Figure 3A shown the ori and RepA-N sequence variation between PLE1,4,5 on the one hand, and PLE2,3 on the other. Supplement 1 only shows the alignment of the conserved c-terminus. Can you please add the N-terminal sequence to enable more detailed examination of the level of sequence variation / conservation between the different groups?

We have expanded the supplementary figure to incorporate the full coding sequence.

4. The similarity of the nuclease domain to the putative homing endonucleases in ICP1 is intriguing, and provides a plausible explanation for the origin of the nuclease domain. How common are these homing endonucleases? Can the authors make a tree of these nuclease domains to see if this Odn nuclease domain fall inside the tree? Or do you not have enough sequences for this / is sequence divergence too high?

Homing endonuclease are fairly common within lytic bacteriophages though the extent of this varies between phage clades. This family of T5orf172 homing endonucleases appear to be fairly common in ICP1 related phages, though only four HEGs in ICP1^2001^ show noticeable similarity to Odn’s nuclease domain. This family of HEGs is discussed in a recent publication about PLE and ICP1 (Netter et al.), a more detailed discussion of ICP1 HEGs and their potential role in ICP1 genome evolution will appear in a forthcoming review (Boyd et al. Annual Reviews Virology in press), and we’re continuing work to understand the relationship between different HEGs in ICP1 and related phages.

As requested, we’ve added a tree of the aligned HEG domains (Figure 4—figure supplement 1). It’s a small number of sequences that are short in length (~200aa), but we think the case for horizontal transfer of the Odn endonuclease domain is compelling.

5. Figure 4 D – nice to see the cleavage activity of gp88/Odn. It should not be too hard to identify the cleavage site using Sanger sequencing (it looks like sharp bands, so i assume a single cut). Have you tried this?

We have! Please see our response to essential revisions point 6.

6. Line 456-459 – I don't think the data are sufficient to draw this conclusion; even if the mutation provides resistance to Odn, it is impossible to work out from these data whether or not the Ori mutations are adaptive, and if so, whether or not this is because of ICP1 infections. This statement therefore needs to be further toned down in my opinion.

Thank you for this suggestion. After reviewing what we had written, we agree that the original statement was over reaching. We’ve amended this passage to read:

“These data demonstrate that PLE can escape Odn activity through subtle restructuring of the iterons, in addition to more extensive replication module exchange (Figure 3A), and suggest that ICP1 defenses like Odn may select for diversification of the PLE replication machinery.”

7. Line 544 – the Gomez et al. Science paper from the same group would also be appropriate to cite.

Thank you for this suggestion, we have added this citation.